# The impact of changing the land surface scheme in ACCESS(v1.0/1.1) on the surface climatology

Eva A. Kowalczyk[1], Lauren E. Stevens[1], Rachel M. Law[1], Ian N. Harman[2], Martin Dix[1], Charmaine N. Franklin[1], and Ying-Ping Wang[1]

[1]CSIRO Oceans and Atmosphere, Aspendale, Victoria 3195, Australia
[2]CSIRO Oceans and Atmosphere, Yarralumla, ACT, 2600, Australia

*Correspondence to:* Eva Kowalczyk (eva.kowalczyk@csiro.au)

**Abstract.** The Community Atmosphere Biosphere Land Exchange (CABLE) model has been coupled to the UK Met Office Unified Model (UM) within the existing framework of the Australian Community Climate and Earth System Simulator (AC-CESS), replacing the Met Office Surface Exchange Scheme (MOSES). Here we investigate how features of the CABLE model impact on present day surface climate using ACCESS atmosphere-only simulations. The main differences attributed to CABLE include a warmer winter and a cooler summer in the Northern Hemisphere (NH), earlier NH spring runoff from snowmelt, and smaller seasonal and diurnal temperature ranges. The cooler NH summer temperatures in canopy covered regions are more consistent with observations and are attributed to two factors. Firstly CABLE accounts for aerodynamic and radiative interactions between the canopy and the ground below; this placement of the canopy above the ground eliminates the need for a separate bare ground tile in canopy covered areas. Secondly, CABLE simulates larger evapotranspiration fluxes and slightly larger daytime cloud cover fraction. Warmer NH winter temperatures result from the parameterization of cold climate processes in CABLE in snow covered areas. In particular prognostic snow density increases through the winter and lowers the diurnally resolved snow albedo; variable snow thermal conductivity prevents early winter heat loss but allows more heat to enter the ground as the snow season progresses; liquid precipitation freezing within the snowpack delays the building of the snow pack in autumn and accelerates snow melting in spring. Overall we find that the ACCESS simulation of surface air temperature benefits from the specific representation of the turbulent transport within and just above the canopy in the roughness sublayer as well as the more complex snow scheme in CABLE relative to MOSES.

## 1   Introduction

One of the main issues in climate modelling is understanding the dependence of climate on the interaction between clouds, radiation, precipitation and the land surface processes. A land surface model (LSM) is one of the key components of a climate model, providing information on surface exchange processes. The LSM includes a representation of the turbulent transport of momentum, heat and water between the land surface, canopy and the atmospheric boundary layer, as well as descriptions of thermal and hydrological processes in the soil and snow. A number of studies have been conducted to understand land-atmosphere interactions. Betts (2009) synthesised 15 years of his published work discussing the basic physical processes

involved in the land-surface-atmosphere interactions as well as their relationships from the modelling and observational perspective. The paper discussed the role of the surface and cloud albedo on radiation and surface fluxes, the role of soil water availability and clouds on the partitioning of the surface energy and the diurnal cycle of temperature, the role of soil moisture in evaporation-precipitation feedback, and the role of surface and atmospheric processes in determining Boundary Layer

equilibrium. Betts (2009) examined systematic features of the seasonal and diurnal cycles as well as the coupling of processes and compared their observable relationships with their model simulations. The feedbacks between soil moisture and climate was examined in Koster et al. (2004) where multimodel experiment identified/estimated regions where precipitation is affected by soil moisture anomalies during Northern Hemisphere summer. The interaction between soil moisture and precipitation is complex as it has direct and indirect effects. Direct effect such as moisture recycling is described in Eltahir and Bras (1996).

Indirect effects including the influence of soil moisture on boundary layer and clouds are investigated in Ek and Holtslag (2004) and Taylor et al. (2011). The effect of land surface processes on extreme events was described in Seneviratne et al. (2010). Fischer et al. (2007) shows that more than half of the summer heat waves in Europe have contributions from soil moisture and temperature interactions. The effect of dry soils in southern Europe on summertime heat waves and drought were described in Vautard et al. (2007) and Zampieri et al. (2009). Hirsch et al. (2014) identified that soil moisture-temperature feedbacks were

affecting daily maximum temperature in Australia. Feedbacks from climate change that generate variations in soil moisture are described in Seneviratne et al. (2013), Berg et al. (2016) and Lorenz et al. (2016). Berg et al. (2016) showed that the aridity response is amplified by land-atmosphere feedbacks under global warming.

With rapidly increasing changes in land management and land use producing complex feedbacks between the biosphere and climate, LSMs have become increasingly complex. The performance of different LSMs has been compared using pre-

scribed meteorological forcing (e.g. Slater et al., 2001; Luo et al., 2003; Abramowitz et al., 2008; Best et al., 2015) and benchmarking systems for land surface models are being developed (Abramowitz, 2012; Kumar et al., 2012; Luo et al., 2012). Comparisons of different land surface models within a single atmospheric model are less common, due to the coupling work involved, although tools are being developed to provide a standard coupling interface (e.g. NASA's Land Information System, http://lis.gsfc.nasa.gov/, Kumar et al., 2006). Here we explore the impact on the simulated climate by changing the LSM in

an atmospheric model (the UM) from the original scheme that was developed with the model (MOSES) to an alternate LSM (CABLE).

The comparison of these LSMs is part of the development of ACCESS, used for both numerical weather prediction (NWP) (Puri et al., 2013) and climate modelling (Bi et al., 2013), with the LSM evaluation currently focussed on the climate timescale with evaluations at NWP timescales to follow. Two ACCESS versions contributed to the 5th Coupled Model Intercomparison

Project (CMIP5) using the two different LSMs, MOSES and CABLE. However evaluation of the impact of the LSM was complicated by other differences in the atmospheric settings and cloud scheme between the two versions. Thus while Kowalczyk et al. (2013), (hereafter referred to as K2013), noted significant differences in the simulated seasonal and diurnal temperature ranges and in timing of runoff from snowmelt in the Northern Hemisphere from the different ACCESS versions, these could not be attributed solely to the LSM used. Hence we aim to clarify that attribution by performing present-day atmosphere-only

simulations with model versions that only differ in their choice of LSM. A second aim is to explore which processes within the

LSMs are driving the differences and where differences in process representation (Sec. 2.2) between the LSMs appears to be important.

We investigate the diurnal cycle as well as mean seasonal and annual time scales of near surface meteorological variables. Simulation of the phase and amplitude of the diurnal cycle of the near surface variables allows the testing of the model representation of the interaction between the surface, boundary layer and the atmosphere above. A focus on summer (Sec. 4.2) and winter (Sec. 4.3) separately highlights the different processes that are important in different seasons.

## 2  The ACCESS model

The atmospheric component of ACCESS (Bi et al., 2013) used in these simulations is the UK Met Office UM with HadGEM2(r1.1) atmospheric physics as described in Davies et al. (2005) and The HadGEM2 Development Team: et al. (2011). Two versions of ACCESS are used here: ACCESS1.0 uses the original UM LSM, MOSES, and was one of the ACCESS versions submitted to CMIP5, ACCESS1.1 replaces MOSES with CABLE v1.8 (Kowalczyk et al., 2006; Wang et al., 2011) but otherwise leaves the atmospheric model unchanged. This study will focus on the comparison between ACCESS1.0 and ACCESS1.1. The evaluation will, however help interpret results from ACCESS1.3 (Bi et al., 2013; Kowalczyk et al., 2013), the alternate ACCESS version used for CMIP5 which used CABLE and different atmospheric settings.

### 2.1  Land surface model descriptions

Land surface models CABLE and MOSES include mechanistic formulations of the physical, biophysical and biogeochemical processes that control the exchange of momentum, radiation, heat, water and carbon fluxes between the land surface and the atmosphere. Both models use tiles to represent land cover types in each grid-cell and calculate a separate energy balance for each tile to provide area-weighted grid mean fluxes and temperatures.

A basic configuration of MOSES version 2.2 was used in the ACCESS1.0 simulation (Cox et al., 1999; Essery et al., 2001) and is also used for ACCESS numerical weather prediction. The MOSES code formed the scientific core of the Joint UK Land Environment Simulator (JULES) (Best et al., 2011), which has both stand-alone and Unified Model (UM) implementations and has had ongoing development since the version of MOSES used here. In MOSES, the canopy is modelled as one big leaf model and is represented in the surface energy balance equation through the coupling to the soil underneath. The soil underneath is not tiled and hence a homogenous soil moisture and temperature is common to all tiles within a grid cell. Subsurface tiling is used in CABLE.

The CABLE model (v1.8) has been coupled to the UM and is used in ACCESS1.1 simulations. CABLE is a one layer two-leaf canopy model, as described in Wang and Leuning (1998) and was formulated on the basis of the multilayer model of Leuning (1995). CABLEv1.8 is derived from CABLEv1.4b (Kowalczyk et al., 2006; Wang et al., 2011) with the changes for CABLEv1.8 detailed in K2013.

## 2.2 Differences between CABLE and MOSES

The main difference between CABLE and MOSES is the representation of the canopy processes including the structural placement of the canopy above the bare ground; there are also significant differences in snow submodels (Table 1). In MOSES a "two-patch" approach is used in which the canopy is modelled by conceptually placing it beside bare ground and calculating

entirely separate energy balances for bare ground and vegetation, hence neglecting radiative and aerodynamic interaction between the two systems and their mediation of each others microclimate. Figure 1a gives an example of a mean grid-cell flux density calculation i.e. sensible heat flux is calculated from the weighted fraction of the vegetation fraction tile ($\sigma$) and the bare ground tile ($1 - \sigma$).

Surface temperature, $T_r$ (K), in MOSES is interpreted as a surface skin temperature (Essery et al., 2001) and is obtained for

both vegetation and bare ground tile from the surface energy balance calculated as

$$C_s \frac{dT_r}{dt} = R_n - H - LE - G_0 \qquad (1)$$

where $R_n$ is surface net radiation ($\mathrm{Wm}^{-2}$), H is the sensible heat flux ($\mathrm{Wm}^{-2}$), LE is the latent heat flux ($\mathrm{Wm}^{-2}$), where L is the latent heat of vaporization ($\mathrm{Jkg}^{-1}$) and E is the evaporation ($\mathrm{kg\,m}^{-2}$). $C_s$ is a volumetric heat capacity calculated as the weighted sum of the heat capacity of dry soil, liquid and ice ($\mathrm{JK}^{-1}\mathrm{m}^{-2}$) and $G_0$ is the heat flux ($\mathrm{Wm}^{-2}$) into the ground

parameterised as

$$G_0 = f_r(\sigma T_r^4 - \sigma T_s^4) + (1 - f_r)\frac{2c}{\Delta z_s}(T_r - T_s) \qquad (2)$$

where $\Delta z_s$ and $T_s$ are the thickness (m) and temperature (K) of the top soil layer respectively, $f_r$ is radiative canopy fraction ($f_r{=}1{-}e^{LAI/2}$), $\sigma = 5.67 \times 10^{-8} \ \mathrm{Wm}^{-2}\mathrm{K}^{-4}$ is the Stefan Boltzmann constant and c is the thermal conductivity ($\mathrm{Wm}^{-1}\mathrm{K}^{-1}$). Components of net radiation ($R_n$): incoming long wave ($\mathrm{Wm}^{-2}$) and the net short wave ($\mathrm{Wm}^{-2}$) are calculated outside of the

LSM by the UM atmospheric radiation model. The heat diffusion equation is solved to calculate the soil temperature, $T_s$ (K).

By contrast, in CABLE the canopy is placed conceptually above the ground (Fig. 1b) hence removing a need for a separate bare ground tile in canopy covered areas (Fig. 2). A combined energy balance for the soil-vegetation system is calculated allowing for the aerodynamic and radiative interaction between the canopy and the ground (Kowalczyk et al., 2006). The mean grid flux density is a sum of the soil flux and the canopy flux (Fig. 1b). When solving the combined energy balance,

the calculation of surface fluxes depends on stability and the surface temperature and simultaneously the surface temperature depends on the stability and fluxes. Therefore, an iterative procedure is used to allow for the simultaneous calculation of all the required variables. We first calculate the radiation absorbed by the canopy, differentiating between sunlit and shaded leaves. We iterate for the thermal stability parameter and soil heat fluxes simultaneously with the solution of the coupled model of stomatal conductance calculating photosynthesis, heat fluxes, leaf ($T_l$) and vegetation temperatures, $T_v$ (K). At this stage the

soil surface temperature from the previous time step is being used in the iteration. Having obtained canopy/soil fluxes and canopy temperature, the heat flux into the ground is obtained by

$$G_0 = S_{abs} + L_{in} + (1 - \tau)\epsilon_l \sigma T_v^4 - \epsilon_s \sigma T_s^4 - H_s - LE_s \tag{3}$$

where $S_{abs}$ is net short wave at the soil surface ($Wm^{-2}$), $L_{in}$ is incoming long wave ($Wm^{-2}$) which includes terrestrial and canopy irradiances. $T_s$ (K) is the soil surface temperature which in CABLE is the temperature of the top thin soil layer of 0.022 m. The heat diffusion equation is solved to calculate the soil temperature profile. $\epsilon_l$ and $\epsilon_s$ are leaf and soil emissivity and $H_s$ and $LE_s$ are soil heat fluxes ($Wm^{-2}$). The surface radiative temperature ($T_r$) is obtained from vegetation $T_v$ (K) and soil surface temperatures;

$$T_r = ((1 - \tau)T_v^4 + \tau T_s^4)^{1/4} \tag{4}$$

where $\tau$ is a canopy transmission $\tau$ = exp(-c LAI), c is an extinction coefficent for beam radiation and black leaves (Wang and Leuning, 1998, Eq. B6).

CABLE has a more complex representation of canopy turbulent transport than many other land surface models which use conventional rough wall boundary theory. In particular, features of the canopy representation in CABLE that are not present in MOSES are:

- Turbulent transport within the canopy based on Localised Near-Field theory, (Raupach, 1989) and transport just above the canopy in the roughness sublayer (RSL) is simulated. The inclusion of a representation of the RSL is critical to the performance of CABLE.

- The model differentiates between sunlit and shaded leaves for the calculation of canopy radiation, photosynthesis, stomatal conductance and leaf temperature (Wang and Leuning, 1998).

- The canopy albedo is resolved diurnally as a function of beam fraction, the sun angle, canopy leaf area index, leaf angle distribution and the transmittance and reflectance of the leaves.

In CABLE the two main canopy parameters affecting turbulent exchange i.e. the displacement height, d, and the roughness length for momentum, $z0_c$, have more complex representation than many other LSMs where these parameters are a constant fraction of canopy height. The displacement height, which describes the mean level of momentum absorption by the canopy, is a function of canopy height and leaf area index as given in Raupach (1994) (Eq. 8). The canopy roughness length, $z0_c$, is determined by matching the mean wind speed profiles within and above the canopy as described in Raupach (1994). In MOSES, a more conventional rough wall boundary theory is used, with roughness length being a constant fraction of the canopy height (h) i.e. $z0_c = h/20$ for trees and h/10 for other vegetation types. Displacement height is not explicitly included in its formulation, with the result being that the reference level for wind is the height above the displacement height for each tile and consequently

the ground surface is uneven. Both models use the same prescribed value of soil roughness length, $z0_{soil} = 3 \times 10^{-4}$, as well as a common geographically explicit snow free soil albedo dataset. A more recent version of CABLE than used here allows soil albedo to be calculated from soil moisture and colour (Kala et al., 2014) but has only been applied to offline CABLE simulations of the Australian continent.

Both LSMs use multiple surface types in each grid-cell, but with different numbers of vegetated and non-vegetated types (Sec. 3.1). Subsurface tiling is used in CABLE, where each surface tile has a corresponding soil tile for the calculation of soil temperature, moisture and runoff, while in MOSES the soil is common to all tiles within a grid cell. Soil processes are modelled similarly in both models but with different vertical resolution. Soil temperature and moisture are calculated for four soil layers to a depth of 3.0m in MOSES and six layers to a depth of 4.6m in CABLE. In both models soil moisture is calculated using

Richards' equation. The evolution of soil moisture depends on the rates of infiltration, plant transpiration, soil evaporation and deep drainage. The heat diffusion equation, including an explicit freeze-thaw scheme, is solved to calculate the soil temperature profile. In CABLE soil water is assumed to be at the ground temperature so there is no heat exchange between the soil moisture and the soil due to vertical movement of water. MOSES calculates the advection of heat by moisture fluxes.

There are also significant differences between snow model components used in ACCESS1.0 and ACCESS1.1 simulations.

In both models snow cover evolution is based on the mass budget between the snowfall, sublimation and the snowmelt. The amount of snow deposited on the surface depends on the amount of solid/liquid precipitation which in the UM is computed by the cloud microphysics parameterizations. Both models accumulate solid fraction at the snowpack surface but the treatment of liquid fraction is different. In CABLE rain falling on snow freezes within the snowpack while MOSES diverts the rainfall straight to runoff.

The total surface albedo is calculated from the contribution from vegetation, snow and bare ground, the last one being the same in both models. In the version of MOSES used here the albedo for soil, vegetation, ice and snow are specified as single values for all radiation bands. The snow albedo in MOSES remains constant when the surface air temperature is below -2$^{\circ}$C, and undergoes an aging process decreasing its value above -2$^{\circ}$C, see Essery et al. (2001). In CABLE only snow free soil albedo is prescribed. The canopy albedo is resolved diurnally while the snow albedo depends on snow depth, a spectral mix of

the incident solar radiation, soot loading, snow melting/freezing and snow age, which is parameterized as a function of snow density, see Dickinson et al. (1993).

In CABLE, the snow metamorphism and the bulk snow properties are accounted for through changes in snow density, see Gordon et al. (2002). In CABLE the density of the fresh snow is 120 kg/m$^3$, and with time it may increase to 400 kg/m$^3$ while in MOSES it remains constant at 250 kg/m$^3$. Snow density affects the temperature of the snow through its effects on the

snow albedo and thermal conductivity. In CABLE, thermal conductivity for new snow is 0.2 Wm$^{-2}$K$^{-1}$ and increases with snow density up to 0.5 Wm$^{-2}$K$^{-1}$ while in MOSES it remains constant at 0.265 Wm$^{-2}$K$^{-1}$.

## 3 Data and model setup

### 3.1 Model Datasets

The simulation results of MOSES and CABLE also depend on the values of their parameters, with some vegetation or soil type dependent, and others having an explicit geographical distribution. Both models use a number of surface datasets to derive the distributions of vegetation and soil types as well as some of the parameters required for the vegetation and soil, see K2013.

Both ACCESS1.0 and ACCESS1.1 use the IGBP (Global Soil Data Task Group, 2000) soil data. The hydraulic properties are determined from information on soil texture based on empirical relationships (Jones, 2008). Each soil type is described by the following hydraulic characteristics: volumetric water content at saturation, wilting point, field capacity, hydraulic conductivity and matrix potential at saturation. These properties define soil water holding capacity and control the rate of water infiltration through the soil. Soil thermal conductivity and heat capacity depend on soil moisture and ice content.

Both CABLE and MOSES use the same spatially varying snow free soil albedo data set which was obtained by blending soil albedo from Wilson and Henderson-Sellers (1985) with MODIS-derived albedo as described in Houldcroft et al. (2009), for details see Jones (2008).

ACCESS1.0 with MOSES uses five vegetated surface types (broadleaf trees, needleleaf trees, C3 grass, C4 grass and shrubs) and four non-vegetated types (urban, inland water, bare soil and ice). The spatial distribution of surface types is derived from $0.5^{\circ}$ by $0.5^{\circ}$ International Geosphere Biosphere Program (IGBP) data (Loveland et al., 2000). The implementation of CABLE in ACCESS1.1 uses ten vegetated surface types and three non-vegetated types. A dataset prepared for the Common Land Model 4 (CLM4) (Lawrence et al., 2012) at $0.5^{\circ}$ by $0.5^{\circ}$ resolution was mapped to CABLE vegetated types; and wetlands, lakes and permanent ice where taken from IGBP and do not change in time (K2013). Here we use a maximum of 5 tiles per grid cell but CABLE is flexible in the number of tiles used. The vegetation distribution used for both models in this study is for present-day conditions, i.e. 2005. Figure 2 in K2013 shows the differences in vegetation fractions. In general the distributions are broadly similar for both models. The main difference is in the representation of bare ground underneath a canopy, shown here in Fig. 2. CABLE's vegetation is above the ground hence there are many grids-cells in CABLE without bare ground tile (Fig. 2b). By contrast, MOSES' vegetation is placed beside bare ground, and hence every grid-cell is allocated a separate bare ground tile to account for bare ground under a canopy (Fig. 2a). The vertical placement of the vegetation above the ground also has implications on the calculation of the surface albedo and roughness length which in CABLE are the integral part of the model.

The key parameters for each CABLE surface type used in the simulation are given in K2013. A description of vegetation parameters used by MOSES can be found in Cox et al. (1999), Cox (2001) and Jones (2008). MOSES uses prescribed monthly varying Leaf Area Index (LAI) which depends on vegetation type and canopy height. In the ACCESS1.1 simulations described here, LAI is prescribed from MODIS satellite estimates (Yang et al., 2006). However, unlike MOSES, a constant value is used across all tiles within a grid-cell. This consequently limits the differentiation of vegetated surfaces within a grid-cell, a limitation that needs to be addressed in future implementations of CABLE in ACCESS.

Both MOSES and CABLE are able to predict canopy LAI when coupled to appropriate submodels that simulate plant growth: TRIFFID (Cox, 2001) for MOSES, CASA-CNP (Wang et al., 2010) for CABLE. However these submodels are not used in the ACCESS simulations described here.

## 3.2 Observations

Both global and site-based datasets have been used to provide an observational context to the comparison between model versions. For evaluating regional to continental scale model differences, we use the ERA Interim Reanalysis product (ERAi, Dee et al., 2011), the Global Precipitation Climatology Centre (GPCC) monthly version 7 precipitation dataset (Becker et al., 2013; Schneider et al., 2015), the International Satellite Cloud Climatology Project (ISCCP) D2 data product (http://isccp.giss.nasa.gov/products/c Rossow and Schiffer, 1991; Rossow et al., 1996), and the CRU3.22 near-surface land temperature dataset (University of East Anglia Climatic Research Unit et al., 2014). FLUXNET flux station data (Baldocchi et al., 2001) are used for more detailed site-based analyses, to help identify which processes contribute to generating regional differences in the model simulations.

## 3.3 Model setup

We perform atmosphere-only simulations following the Atmosphere Model Intercomparison Project (AMIP) experimental design with prescribed sea surface temperature and sea-ice to constrain the climatology and aid in interpreting the differences between ACCESS1.0 (MOSES) and ACCESS1.1 (CABLE) simulations. We run both models for 20 years for the period 1979-1998 at a resolution of $1.875^{\circ} \times 1.25^{\circ}$ (N96). The same atmospheric model and cloud scheme are used in ACCESS1.0 and ACCESS1.1. Similar to a previous study by K2013, both simulations use initial conditions from a pre-industrial simulation. Global atmospheric $CO_2$ is also prescribed, increasing from 337 ppm in 1979 to 379 ppm in 2005, although this increase is not passed to CABLE (in ACCESS1.1), which uses a constant 370 ppm in this implementation. An additional sensitivity experiment (ACCESS1.0L) was also performed, using ACCESS1.0 (MOSES) but with LAI taken from the ACCESS1.1 (CABLE) case and, as in ACCESS1.1, using the same LAI for all tiles within a grid-cell. We also make reference to the 27 year (1979-2005) ACCESS1.3 AMIP simulation submitted to CMIP5, which employed CABLE1.8 but with additional changes to the atmospheric physics parameterisations (configuration similar to Hewitt et al. (2011), Appendix A).

We also perform single-site offline simulations to explain some of the implications of different processes between models. Note that the offline models are not identical to that used in the ACCESS simulations (for MOSES we use JULES v3.0 and for CABLE we use v2.1.2), as earlier versions of the code were not setup to easily switch between offline and online simulations. However, the core science parameterisations are essentially the same between the online and offline versions of the model used in this study. Hence the aim is not to exactly reproduce the online behaviour but rather to characterise differences in model behaviour when using common meteorological forcing.

## 4 Model results

We focus our assessment of the land surface climatology on the seasonal means of screen level temperature for present-day conditions. We calculate means for December-January-February (DJF) and June-July-August (JJA) for 1979-1998. After a brief comparison with observations and ERAi reanalysis, we seek a process-based understanding of the differences in NH land temperature between the two model simulations.

### 4.1 Mean climate

Modelling climate over the land is critically dependent on the interaction between clouds and the surface. Clouds are precursors of precipitation, reflect solar radiation and absorb outgoing long-wave radiation affecting the surface energy balance. Figure 3a shows the zonally averaged simulated total cloud cover fraction over land in comparison with ERAi derived cloud fraction and ISCCP observations. Both ACCESS1.0 and ACCESS1.1 produce much smaller cloud fractions than ACCESS1.3, especially in the tropics and the polar regions, illustrating the large impact of changing the atmospheric physics settings and cloud scheme in ACCESS1.3 (K2013). In comparison with ACCESS1.0, ACCESS1.1 simulates slightly larger cloud fractions around the equator, in mid-high latitudes in the Northern hemisphere and in the southern polar regions. However, in comparison with ERAi and ISCCP, both ACCESS1.0 and ACCESS1.1 underestimate cloud fraction in the tropics (while ACCESS1.3 is a better fit to the observations and reanalysis). Around 30° N and S, ERAi and ISCCP tend to span the model simulations while in polar regions ACCESS1.0/1.1 are closer to the reanalysis and observations than ACCESS1.3.

Zonally averaged land-only mean precipitation (Fig. 3b) is similar in ACCESS1.0 and ACCESS1.1 and lower than ACCESS1.3 in the tropics. Consistent with the cloud cover, land precipitation in the tropics is underestimated in ACCESS1.0 and ACCESS1.1 compared with both ERAi and GPCC. By contrast, in the northern mid-latitudes, the simulations give slightly more precipitation than observed. Table 2 presents model computed and 'observed/estimated' components of the water balance over the global land area. The estimates come from Baumgartner and Reichel (1975) and Legates and Willmott (1990). Globally, both ACCESS1.1 and ACCESS1.0 produced similar means for precipitation and evapotranspiration but larger differences are found for boreal summer over the Northern mid-high latitudes (Table 3). This is consistent with the larger cloud fraction simulated in these areas by ACCESS1.1 (Fig. 3a).

Mean screen temperature biases relative to ERAi (Fig. 4) are similar for ACCESS1.0 and ACCESS1.1, at least across the tropics and more generally in DJF. The significance of these biases (indicated by shading in Fig. 4) has been assessed using a modified t-test (Zwiers and von Storch, 1995) with a significance level of 0.05; this test accounts for auto-correlation and we use the look up table method to allow for the relatively small sample size. The tropical biases tend to be significant while in the northern mid-high latitudes the significance varies with season, region and model. In DJF significant cold biases cover a larger fraction of the northern mid-high latitudes in ACCESS1.0 than ACCESS1.1, while ACCESS1.1 shows small regions of significant positive bias. In JJA common (though larger and regionally more significant in ACCESS1.0) warm biases occur across central Europe and the Great Plains of North America, coincident with the underestimation of precipitation in these regions (not shown but very similar to K2013, Fig. 4b). Likewise, significant warm temperature biases in the Indian

peninsula, equatorial Africa and part of the Amazon also result from the underestimation in rainfall, enhanced further by a positive feedback between the decrease in evapotranspiration and increased solar radiation due to a deficit in cloud cover fraction (Fig. 3a). We note, however that these warm biases are smaller and significant less often when compared with the CRU temperatures rather than ERAi (not shown), especially for the Amazon region. The warm biases which are specific to

ACCESS1.0 include those in the high and mid latitudes of Asia and high latitudes of North America in JJA and unexpectedly, a strong bias over Antarctica in DJF. The ACCESS1.1 simulation tends to have a warm bias in some mountainous snow covered regions. For example, in East Siberia in DJF (and larger relative to CRU than ERAi) and Antarctica (in JJA); where the mean winter temperature drops below -20$^{\circ}$C, ACCESS1.1 overestimates the daily temperature by up to 5$^{\circ}$C over areas of high topography. There are also common and significant cold biases occurring over arid areas of North Africa and the Middle East,

with the biases generally slightly larger for ACCESS1.1. Overall, for the northern mid-latitudes, ACCESS1.1 gives smaller biases relative to ERAi than ACCESS1.0 while in DJF it is not clear that one simulation is less biased than the other.

Table 2 summarises annual mean, minimum and maximum temperature for all land and excluding Antarctica, and energy budget components with estimates from Henning (1989), Budyko (1978) and Smith et al. (2008). Excluding the Antarctic continent, where the largest temperature differences occur, ACCESS1.1 simulates a cooler mean screen temperature by 0.5$^{\circ}$C,

dominated by the difference in maximum temperature (lower by 0.86$^{\circ}$C) with the minimum temperature slightly higher (by 0.09$^{\circ}$C). Over Northern land (30-90$^{\circ}$N) (Table 3), ACCESS1.1 is cooler by about 0.5$^{\circ}$C, with mean maximum temperature cooler by about 1.3$^{\circ}$C and the minimum temperature warmer by 0.4$^{\circ}$C. Seasonal temperature differences are larger; in boreal winter the ACCESS1.1 minimum temperature was warmer by 1.7$^{\circ}$C while in summer the ACCESS1.1 maximum temperature was cooler by 2.7$^{\circ}$C.

In the comparison below we will focus on the Northern hemisphere, where the surface air temperature shows significant differences between both simulations (Fig. 4e,f). ACCESS1.1 is generally warmer than ACCESS1.0 in DJF but the significant differences are mostly confined to the high altitude regions of Asia. In JJA ACCESS1.1 is cooler than ACCESS1.0 and the significant differences are more widespread. We separately discuss boreal summer and winter, as the cold season with surface snow has a mostly stable boundary layer in contrast to the warm season which has an unstable daytime boundary layer.

**4.2   Boreal summer**

In JJA, Northern Hemisphere canopy-covered areas show mean screen level temperatures that are lower in ACCESS1.1 (with CABLE) by up to several degrees (Fig. 4f), despite ACCESS1.1 simulating lower surface albedo (Fig. 5b). The relative cooling is larger and more widespread for maximum temperature than minimum temperatures (Fig. 6b,d). The 2.1$^{\circ}$C difference in northern continental JJA temperature between model simulations (Table 3) is larger than the interannual variability in either

simulation (standard deviation=0.4-0.5$^{\circ}$C), with the interannual variability being moderately well correlated between the two simulations (R$^2$=0.7). This suggests the JJA temperature difference between ACCESS1.0 and ACCESS1.1 is robust. There are two main reasons for these differences: the first one is each model's approach to canopy representation i.e. the "two-patch" approach conceptually placing canopy beside bare ground in MOSES compared to above the ground in CABLE (Fig. 1). The second are feedbacks enhancing the precipitation due to larger evaporation fluxes. Differences in LAI between ACCESS1.0 and

ACCESS1.1 do not make a major contribution to the differences in temperature. The LAI sensitivity simulation (ACCESS1.0L) gives northern continental JJA temperature (mean = 19.22) much closer to ACCESS1.0 (root mean square difference of 0.4 K)) than to ACCESS1.1 (RMS difference of 2.7 K), indicating that a change in LAI has not significantly changed the ACCESS1.0 simulation of the northern continent in summer.

To illustrate the impact of canopy representation, we show an offline simulation for a single location, a 15 m tall Scots pine forest at Hyytiälä (61.85°N, 24.3°E) for 2002-2005. This site is represented in CABLE as a single tile with evergreen needleleaf vegetation above the ground while in JULES (based on MOSES) the site is represented with two tiles; a needleleaf canopy (tile fraction of 0.8) and bare ground (0.2). JULES's calculated midday net radiation (320 $\mathrm{W\,m^{-2}}$) is similar for both tiles with the needleleaf tile having a slightly larger value due to the lower vegetation albedo. However, in CABLE, net radiation

for the canopy reaches a midday value of 290 $\mathrm{W\,m^{-2}}$ with 90 $\mathrm{W\,m^{-2}}$ for the bare ground underneath, adding to a total grid maximum flux around midday of 380 $\mathrm{W\,m^{-2}}$. The canopy temperature in CABLE and canopy tile temperature in JULES have similar diurnal variation and amplitudes; however the midday ground surface temperature in CABLE is 6°C cooler than the bare ground tile temperature in JULES (Fig. 7) since CABLE's soil is shaded by the canopy while JULES' bare ground tile is exposed to the full atmospheric forcing. In July LAI is about 2.4 at this site resulting in a low canopy transmission coefficient

and a mean grid radiative temperature in CABLE (Eq. 4) that is close to the canopy temperature, with only a slight reduction in midday temperature due to the lower soil temperature. By contrast, the averaging of bare ground and vegetated tiles in JULES leads to a midday grid temperature slightly higher than that obtained for the vegetated tile alone. The consequence is that JULES is warmer by up to 1.5°C at midday. In ACCESS1.1 large areas of the globe do not have a bare ground tile while in ACCESS1.0 up to 20% of the grid-cell fraction in canopy-covered areas is designated as bare ground (Fig. 2). This

representation impacts the overall calculation of the grid surface temperature. In particular, it is well known that seasonal depletion of soil moisture over bare ground is larger than in canopy-covered areas due to an absence of plant physiological control over the evapotranspiration fluxes.

In summer, in the mid and high latitudes, the weather and the climate are driven by large scale synoptic systems, and interactions between clouds, precipitation and the Atmospheric Boundary Layer (ABL). The land surface determines the partitioning

of the available energy and provides the moisture and heat fluxes to the ABL. In these regions with relatively moist soils, the key contribution to the climate from the land surface is evapotranspiration, which depends on soil moisture. Table 3 shows that north of 30°N summer mean evaporation and precipitation are larger in ACCESS1.1 by about 0.2 $\mathrm{mm\,day^{-1}}$. Figure 8a shows that around 60-70°N where soil moisture is in abundance, ACCESS1.1 shows significantly larger cloud fraction over canopy covered areas. Total evapotranspiration (Fig. 8b) is also higher in at least half of these areas. Increased evapotranspiration

influences cloud formation and rainfall which in turn replenishes the soil moisture availability for evapotranspiration (Bierkens et al., 2008) (Fig. 8c,d). However we cannot separate cause and effect here i.e. whether higher evaporation fluxes induced higher cloud cover and precipitation or vice versa. Also, not all clouds produce precipitation as water droplets/ice crystals may remain suspended in the atmosphere until they are converted back into vapour. Also note, that most of the areas with the largest model differences in daily maximum temperature (Fig. 6d) coincide with the areas of largest differences in mean precipitation,

(Fig. 8c).

The links between moisture and temperature presented in Figs. 6 and 8 are explored for a typical mid-latitude grid-cell; online simulations of the diurnal cycle of fluxes, temperatures, cloud cover and precipitation are compared with observations for the grid-cell closest to the Boreas flux tower site (55.88°N, 98.48°W), in North America. Comparing a grid-cell from the model simulations with flux tower observations has limitations due to model resolution (grid area of about 200km x 140km), and model errors in simulating the site meteorology but gives useful information on the overall model performance and differences between the models. In this grid ACCESS1.1 has 3 tiles; needleleaf trees (0.83), grass (0.07) and lakes (0.10) and ACCESS1.0 has 6 tiles; broadleaf and needleleaf trees (0.09, 0.50), grass (0.17), shrubs (0.02), lakes (0.10) and bare ground (0.12). In ACCESS1.1, the cloud cover fraction (Fig. 9a) is slightly larger than in ACCESS1.0 during the daytime and much larger at night. The intense summer rainfall events are not reproduced with precipitation slightly larger for ACCESS1.1 (Fig. 9b). The maximum daily net radiation in ACCESS1.1 (Fig. 9c) is lower by up to 35 $\mathrm{W\,m^{-2}}$ due to the larger cloud cover fraction while at night time the outgoing long wave flux is smaller up to 40 $\mathrm{W\,m^{-2}}$ due to significantly lower surface temperatures. Partitioning of the net radiation is different, with CABLE simulating larger latent than sensible heat (Fig. 9d,e) due to greater moisture availability. The Boreas grid-cell is located within an area where ACCESS1.1 has larger soil moisture and precipitation (by up to 1 $\mathrm{mm\,day^{-1}}$) than ACCESS1.0 (Fig. 8c,d). Smaller daytime net radiation, larger evapotranspiration and the larger grid fraction covered with trees shading the ground in CABLE results in cooler diurnal screen level temperatures (Fig. 9f), with the difference in maximum temperature being larger than for the minimum temperature. However, for the Boreas grid-cell, the MOSES partitioning is closer to that observed at the flux station. This difference in partitioning is also seen when averaged across the northern continents (Table 3) with MOSES producing a sensible to latent heat ratio of 0.7 compared to 0.5 for CABLE.

The large cloud fraction overnight in ACCESS1.1 is due to the presence of fog, shown by the fraction of very low cloud, < 111 m, in Fig. 9a. The radiative cooling of the surface in the stable nocturnal boundary layer causes the overlying air to cool to the dew point temperature, generating saturation and cloud in the lowest model levels. The cooler surface temperatures simulated with CABLE require a smaller amount of radiative cooling before saturation of the overlying air is reached, compared to the case with MOSES. Once the fog has formed, longwave radiation cools the cloud top rather than the surface and drives the cloud layer through the generation of turbulence. The presence of fog increases the incoming longwave radiation at the surface, leading to an increase in the net surface radiation and the larger sensible heat fluxes seen in the early morning in ACCESS1.1 in Fig. 9d. The fog layer dissipates when the surface warms after sunrise. In much of the tundra and taiga regions high levels of humidity, fog and mist are observed in summer (Przybylak, 2003). This is captured well in the ACCESS1.1 simulation, with the occurrence of fog rapidly decreasing with latitude.

Over the desert and semi desert areas of the Middle East both models showed cold biases in the mean temperature (Fig. 4), but differ from each other in their diurnal range. Figure 6 showed that ACCESS1.1 simulates a warmer minimum and cooler maximum temperatures than ACCESS1.0. This is especially noticeable in large parts of Iran and Saudi Arabia in JJA. Most model grids in ACCESS1.1 in these areas are represented by one bare ground tile while ACCESS1.0 may have two tiles (bare ground tile and a small fraction canopy tile) (Fig. 2). For ACCESS1.1 the larger bare ground area results in a slightly higher surface albedo (Fig. 5), which contributes to cooler daytime temperature. There is limited cloud cover and precipitation in these

areas and the latent heat flux is small or negligible. The maximum daytime surface radiative temperatures in both models were similar but the night time temperatures were warmer in ACCESS1.1. The daytime maximum sensible heat flux in MOSES was slightly larger, cooling the surface and providing more heat to the atmosphere, resulting in warmer daytime air temperature. In CABLE smaller daytime sensible heat under similar radiative forcing allowed for larger ground heat flux, which combined with the deeper soil column (4.7 vs. 3 m), allowed a larger heat storage and thus modulating the daily temperature amplitude. In both models the diurnal pattern of sensible heat flux is phase shifted after local midday. This phase shift occurs in the deserts due to the diurnal radiative cycle being not in phase with the soil heat storage cycle.

## 4.3 Boreal winter

During the Boreal winter ACCESS1.1 is warmer than ACCESS1.0 (Fig. 4e), with mean screen level temperature up to several degrees higher in most northern areas where snow occurs. The minimum temperatures are 1.7$^\circ$C warmer and maximum temperatures 0.3$^\circ$C warmer (Table 3). The interannual variability in each simulation is comparable to these differences with the standard deviation of annual minimum and maximum temperatures being around 0.8-1.2$^\circ$C and the correlation between simulations being 0.4-0.5. Thus the winter temperature differences between simulations appear less robust than those in summer.

Snow constitutes a dominant part of the winter environment in mid and high latitudes. It strongly reduces the available energy at the surface through its high reflectivity of solar radiation. The insulating properties of the snow reduce the soil heat to the atmosphere, thus allowing the soil temperature to remain warmer. The surface energy, water budget and seasonal freezing and thawing of the ground are affected by snow processes. Processes of infiltration, soil water transfer and transpiration are suspended upon soil freezing and resume with thawing. During winter LAI is significantly reduced by snow cover and the leaves senescence, and with plant metabolism slowed down vegetation enters a dormant phase. In this phase the impact of vegetation on surface temperature is reduced to an effect of lowering surface albedo in areas where vegetation protrudes through the snow cover. In these environments, the differences between ACCESS1.0 and ACCESS1.1 simulated winter temperatures are attributed to the different representation of the snow processes by the models; these include the parameterization of snow albedo, accumulation, density and thermal conductivity.

The calculated total surface albedo is significantly lower in ACCESS1.1 (Fig. 5a), with the exception of a band of higher albedo stretching from the northern parts of the Scandinavian Peninsula across Russia. This band occurs around the transition from trees to grass and shrubs. In ACCESS1.1 (CABLE), the influence of snow on surface albedo is dependent on LAI, whereas in ACCESS1.0 (MOSES) albedo is influenced by vegetation type. In this transition region, the prescribed LAI in ACCESS1.1 drops to around 0.5, increasing the albedo, while in ACCESS1.0 this region is tree-covered so the ACCESS1.0 albedo remains relatively low. North of this band, the predominant vegetation type is grass/shrubs (see Fig. 2, K2013), causing the ACCESS1.0 albedo to become larger than that of ACCESS1.1. The sensitivity test, ACCESS1.0L, gives very similar albedo results to ACCESS1.0, confirming that the interaction of snow and vegetation in MOSES is driven by vegetation type rather than LAI. In ACCESS1.1 much lower surface albedo occurred in the areas of intermittent snow cover i.e. central USA and central Asia. This difference is due to the later onset of snow cover in autumn and earlier melting. Over the permanent ice,

CABLE's total surface albedo is higher than for MOSES due to a snow albedo refreshing process that allows albedo to remain around its maximum value.

To illustrate how the snow processes differ between the two land surface models and the consequent impacts on the AC-CESS simulations, we have performed offline simulations, forced with observed meteorology, for the 2003-2004 snow season in Hyytiälä. In winter in Hyytiälä, LAI decreases from a summer maximum of 2.85 to 0.71. The widespread lower ACCESS1.1 albedo in winter is reproduced in the offline simulation. For both models the time evolution of surface albedo reflects snow-fall/melting events (Fig. 10a) and CABLE also represents the diurnal variation of snow and canopy albedo on cloud free days. The JULES (MOSES) albedo response to snowfall/snowmelt events is larger than in CABLE as its variation depends only on snow depth and melting temperature. In CABLE the albedo of the surface is affected by overlying canopy albedo as well as snow age and density. In early winter snow albedo in JULES increases more rapidly and remains higher through the rest of the season. During the melting period the surface albedo in JUELS oscillates with the daily temperature variation around -2$^{\circ}$C while in CABLE the albedo decreased rapidly allowing for faster melting of the snow (Fig. 10b).

In CABLE rain falling on snow freezes within the snowpack while JULES/MOSES diverts the rainfall straight to runoff; this results in a deeper snow cover (Fig. 10b) and contributes to warmer snow temperatures (Fig. 10e). In early spring when liquid precipitation frequently occurs, warm rainfall falling on snow accelerates snow melting in CABLE, decreasing the snow albedo. In the ACCESS1.1 simulation there is slightly more snow in the northern part of the continent and less in the south (not shown). This is broadly consistent with more frequent occurrence of liquid precipitation in the south.

Parameterization of snow thermal conductivity and density contribute to a warmer surface temperature in CABLE. In early winter, the snow has low thermal conductivity (0.2) preventing heat loss from the underlying soil. With time, both snow thermal conductivity and density increase (Fig. 10c,d), allowing for more heat to be absorbed by the snow cover and the ground below. The differences in daily mean surface radiative temperatures between the offline simulations are shown in Fig. 10e. In early winter when the snow cover is shallow, the differences tend to be smaller and are related to snowfall/melting events but with time they increase, with maximum differences occurring as melting begins. Consistently, in the ACCESS1.1 simulation variable thermal conductivity and density of snow contribute to warmer mean temperatures and in particular warmer minimum temperatures over the snow areas.

The warmer surface temperature in ACCESS1.1 occurs throughout the diurnal cycle, as can be seen for Boreas, a needle-leaf forest site dominated by snow and frozen soil processes in winter. Figure 11 shows the 20 year mean diurnal cycle for January temperature, fluxes, precipitation and cloud cover fraction. Both models underestimate the temperature in winter but ACCESS1.1 is warmer than ACCESS1.0 by approximately 2$^{\circ}$C (Fig. 11f). Also, the maximum daily screen level temperature occurs an hour or more later in ACCESS1.1 and is closer to the observed. The latent heat is negligible; sensible heat flux is small and underestimated in both models due to underestimated net radiation (Fig. 11d,e). Precipitation is overestimated in both models. Similar behaviour is also seen for grid-cells in Siberia, consistent with widespread warmer temperatures for ACCESS1.1 (Fig. 6a,c).

Parameterization of the cold climate processes in CABLE, which include liquid precipitation freezing within the snowpack, age dependant diurnally resolved snow albedo, prognostic snow density and variable snow thermal conductivity, resulted in

warmer snow surface temperatures than compared to MOSES. ACCESS1.1 mean, maximum and minimum temperatures were warmer than in ACCESS1.0 (Fig. 4 and 6), with the largest difference of $1.74^{\circ}$C in the minimum temperature (Table 3). Northern continental mean winter precipitation, evaporation, runoff and the heat fluxes were similar in both models while net radiation was only slightly larger in CABLE than is the case for MOSES.

One of the consequences of the seasonal temperature difference, between ACCESS1.0 and ACCESS1.1 in the Northern hemisphere high latitudes, is the timing of the calculated snowmelt and runoff. Spring snowmelt is an important source of water to replenish soil water reservoirs, with an excess of water diverted to runoff. In the high latitudes snowmelt is also a source of fresh water to the Arctic sea. An earlier spring and snowmelt affects land-atmosphere carbon exchange, permafrost thaw and ecosystem carbon sequestration in high latitude tundra ecosystems (Li et al., 2014; Humphreys and Lafleur, 2011;
Tang and Zhuang, 2011). Figure 12 shows the difference in mean monthly total runoff generated from the snowmelt. In ACCESS1.1 in spring, the soil moisture in these regions is close to saturation and thus snowmelt flows on the surface along topography as surface runoff. In ACCESS1.0 there is significantly less soil moisture (Fig. 8d), so the snowmelt water enters partially unfrozen soil and seeps slowly through the soil column before emerging months later as drainage from the bottom layer. Hence a substantial amount of runoff is not generated in ACCESS1.0 until June. In ACCESS1.1 the main contribution
to the total runoff in Fig. 12 comes from the surface runoff while in ACCESS1.0 it comes from the drainage. In high latitude regions soil moisture is high because the moisture evaporates slowly and the soil drainage conditions are poor because of the underlying permafrost. These processes are captured in the ACCESS1.1 simulation. Also, the timing of runoff as simulated in ACCESS1.1 is more consistent with the observations from the three main Siberian river watersheds (Yang et al., 2007) than in ACCESS1.0 confirming that the land surface scheme is the main driver of similar differences noted between ACCESS1.0 and
ACCESS1.3 by K2013. Table 3 shows that in JJA the total runoff for ACCESS1.0 is almost twice as large as ACCESS1.1 due to ACCESS1.1 simulating surface runoff from the spring snowmelt in April and May.

## 5  Conclusions

Kowalczyk et al. (2013) highlighted differences in the present day land surface climatology of the two ACCESS submissions to CMIP5, but the impact of the different land surface model used in each simulation was difficult to determine due to other
differences in atmospheric settings. The simulations presented here, using the same atmospheric settings, have allowed the impacts of the land surface model to be determined, with a focus on the processes driving those impacts. Differences found in K2013 that we can now largely attribute to the land surface processes and model configuration include smaller seasonal temperature amplitude manifested by a warmer winter and a cooler summer, and an earlier runoff from snowmelt in the Northern Hemisphere in ACCESS1.1 (CABLE). CABLE also simulates smaller mean diurnal temperature range in JJA and
DJF in most of the areas including sparsely vegetated regions.

During the Boreal summer in the Northern Hemisphere, in spite of the overall lower surface albedo in canopy areas, AC-CESS1.1 is generally cooler over high latitudes. Cooler surface temperatures are attributed to two factors; the first one being the representation of the canopy in CABLE with the vertical placement of the vegetation above the ground which allows for

radiative and aerodynamic interaction between the canopy and the ground below. An offline simulation showed that in CABLE the net available radiation flux at the ground surface below the canopy was much lower than for a separate bare ground tile directly exposed to the atmospheric forcing in MOSES. Hence, the ground temperature in CABLE, being shaded by vegetation, was cooler than the vegetation temperature while in MOSES it is the opposite; day time bare ground tile temperature
was significantly higher than canopy tile temperature. The MOSES configuration of land cover with a separation of the canopy covered grid into bare ground and canopy tile resulted in larger areas of bare ground surface as shown in Fig. 2. A larger area of bare ground exposed directly to the atmosphere contributed to larger diurnal temperature amplitude with a tendency to dry out earlier due to an absence of physiological control over the evaporation flux. Cooler summer temperatures are also attributed to larger soil moisture, precipitation and day time cloud cover fraction in most of the areas in ACCESS1.1. In high latitudes the
low level cloud cover fraction over the canopy covered area at night was higher in ACCESS1.1 due to the presence of fog.

    In winter when vegetation is dormant, warmer temperatures simulated by ACCESS1.1 over the snow covered areas of mid and high latitudes are attributed to differences in the snow parameterization in CABLE compared with MOSES. In particular, CABLE accounts for liquid precipitation freezing within the snow pack, prognostic snow density and variable snow thermal conductivity. Accounting for liquid precipitation freezing within the snowpack delays the build up of the snow pack in autumn
and accelerates snow melting in spring. Snow density is simulated to increase through the winter which lowers the snow albedo and allows for an increased absorption of solar radiation. Variable snow thermal conductivity increases over the snow season, initially preventing heat loss and later allowing more heat to enter the snow/ground.

    One of the deficiencies of the modelled climate in both versions of ACCESS model was the overestimation of evapotranspiration. In some regions this is due to overestimated precipitation caused by continuous but low intensity events in lieu of
less frequent but more intense rainfall which would allow for an increase in the surface runoff and drier soil. The excessive evapotranspiration is a common problem for a number of other models, (Mueller and Seneviratne, 2014). The sensitivity of the parameterisation of stomata opening to the favourable moisture and energy conditions needs to be re-examined in LSMs such as CABLE and MOSES to restrain the evapotranspiration. An alternate parameterisation of stomatal conductance has also been tested in ACCESS (Kala et al., 2015) and tends to reduce evapotranspiration for parts of the northern continents in JJA.
CABLE has a long history of development, originally in CSIRO, and now as an Australian community model. CABLE is widely used in 'standalone' applications, forced with prescribed meteorology and it has also provided the land surface component of a number of Australian climate and air pollution models. With ACCESS now being the primary model in Australia for numerical weather prediction and climate modelling, it has been important to couple CABLE into ACCESS to enable Australian researchers to incorporate their local land surface development work into atmospheric modelling applications. This
study confirms that changing the land surface model in ACCESS from MOSES to CABLE has not degraded the simulation of the present-day seasonal climatology and has generally improved summer temperature biases. The improvement in summer temperatures is due, in part, to the more complex canopy representation in CABLE compared to many other land surface models. Thus ACCESS with CABLE can be confidently used for climate applications, while further work would be required for assessing the performance of ACCESS with CABLE for numerical weather prediction.

**Code availability**

Code availability varies for different components of ACCESS. The UM is licensed by the UK Met Office and is not freely available. JULES is available from https://jules.jchmr.org/software-and-documentation. CABLE is available from https://trac.nci.org.au/svn/cable/. See https://trac.nci.org.au/trac/cable/wiki/CableRegistration for information on registering to use the CA-
5  BLE repository.

*Acknowledgements.* This research was supported by the Australian Government Department of the Environment, the Bureau of Meteorology and CSIRO through the Australian Climate Change Science Programme. This research was undertaken on the NCI National Facility in Canberra, Australia, which is supported by the Australian Commonwealth Government. This work used eddy covariance data acquired by the FLUXNET community and in particular by the following networks: AmeriFlux (U.S. Department of Energy, Biological and Environmental
10  Research, Terrestrial Carbon Program (DE-FG02-04ER63917 and DE-FG02-04ER63911)), AfriFlux, AsiaFlux, CarboAfrica, ChinaFlux, Fluxnet-Canada (supported by CFCAS, NSERC, BIOCAP, Environment Canada, and NRCan), GreenGrass, KoFlux, LBA, NECC, OzFlux, TCOS-Siberia, USCCC. We acknowledge the financial support to the eddy covariance data harmonization provided by CarboEuropeIP, FAO-GTOS-TCO, iLEAPS, Max Planck Institute for Biogeochemistry, National Science Foundation, University of Tuscia, Université Laval and Environment Canada and US Department of Energy and the database development and technical support from Berkeley Water Center,
15  Lawrence Berkeley National Laboratory, Microsoft Research eScience, Oak Ridge National Laboratory, University of California - Berkeley, University of Virginia. The GPCC Precipitation data provided by the NOAA/OAR/ESRL PSD, Boulder, Colorado, USA, from their Web site at http://www.esrl.noaa.gov/psd/.

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

**Table 1.** The list of major differences in structure and canopy, soil and snow components for MOSES as configured in ACCESS1.0 and CABLE as configured in ACCESS1.1.

| Component | MOSES | CABLE |
|---|---|---|
| Canopy | One big leaf model | Two leaf model (sunlit and shaded leaves) |
| | Canopy tile placed besides bare ground tile | Canopy placed above the ground; no need for a separate bare ground tile in canopy areas |
| | Canopy albedo prescribed | Canopy albedo resolved diurnally |
| | | Turbulent transport within the canopy |
| Grid Tiles | 9 surface types (5 vegetated) with up to 9 tiles used in each grid-cell | 13 surface types (10 vegetated) with up to 5 tiles used in each grid-cell |
| Soil | 4 layers, total depth 3m | 6 layers, total depth of 4.6m |
| | No subsurface tiling | Subsurface tiling |
| Snow | 1 layer | 1 layer for shallow snow, 3 layers for deep snow |
| | Liquid precip goes to runoff | Freezes liquid precip within snowpack |
| | Constant desity of 250 kg/m$^3$ | Prognostic snow density; ranges from 120-400 kg/m$^3$ |
| | Constant conductivity of 0.265 Wm$^{-2}$K$^{-1}$ | Variable snow conductivity; ranges from 0.2-0.5 Wm$^{-2}$K$^{-1}$ |
| | Constant snow albedo except when melting | Snow albedo |

**Table 2.** Water and energy budget components, averaged over all land surfaces for ACCESS1.0 and ACCESS1.1 compared to estimates from other sources. (Values in parenthesis are for all land excluding Antarctica).

| | ACCESS1.0 | ACCESS1.1 | Other Estimates |
|---|---|---|---|
| Precipitation (mm/day) | 2.13 (2.30) | 2.19 (2.36) | 2.03[a], 2.05[b] |
| Evaporation (mm/day) | 1.50 (1.64) | 1.54 (1.70) | 1.31[a] |
| Surface Runoff (mm/day) | 0.21 (0.19) | 0.15 (0.12) | 0.73[a] |
| Drainage (mm/day) | 0.51 (0.56) | 0.53 (0.59) | |
| Screen Temperature (°C) | 8.63 (12.98) | 8.08 (12.48) | 8.5[c] |
| - Maximum | 13.33 (17.80) | 12.44 (16.94) | |
| - Minimum | 3.83 (8.08) | 3.85 (8.17) | |
| Sensible Heat (Wm$^{-2}$) | 31.29 (36.18) | 25.46 (29.75) | 30.53[d], 37.31[e] |
| Latent Heat (Wm$^{-2}$) | 43.33 (47.58) | 44.61 (49.14) | 35.86[d], 34.41[e] |
| Net Radiation (Wm$^{-2}$) | 77.51 (86.88) | 72.81 (81.96) | 66.39[d], 72.20[e] |

[a]Baumgartner and Reichel (1975), [b]Legates and Willmott (1990), [c]Smith et al. (2008), [d]Henning (1989), [e]Budyko (1978)

**Table 3.** Water and energy budget components, averaged over all land surfaces above $30^{\circ}$N excluding Greenland - Annual, DJF and JJA.

| | Annual | | DJF | | JJA | |
|---|---|---|---|---|---|---|
| | ACCESS1.0 | ACCESS1.1 | ACCESS1.0 | ACCESS1.1 | ACCESS1.0 | ACCESS1.1 |
| Precipitation (mm/day) | 1.77 | 1.89 | 1.29 | 1.31 | 2.23 | 2.44 |
| Evaporation (mm/day) | 1.26 | 1.36 | 0.36 | 0.40 | 2.36 | 2.54 |
| Surface Runoff (mm/day) | 0.12 | 0.22 | 0.04 | 0.01 | 0.19 | 0.24 |
| Drainage (mm/day) | 0.52 | 0.31 | 0.23 | 0.24 | 0.89 | 0.35 |
| Screen Temperature ($^{\circ}$C) | 2.96 | 2.43 | -13.30 | -12.19 | 19.02 | 16.95 |
| - Maximum | 7.37 | 6.02 | -9.59 | -9.27 | 24.11 | 21.36 |
| - Minimum | -1.53 | -1.16 | -16.73 | -14.99 | 13.40 | 12.30 |
| Sensible Heat (Wm$^{-2}$) | 23.27 | 17.35 | 1.74 | 2.31 | 48.18 | 35.16 |
| Latent Heat (Wm$^{-2}$) | 36.49 | 39.37 | 10.55 | 11.75 | 68.42 | 73.50 |
| Net Radiation (Wm$^{-2}$) | 61.67 | 59.14 | -1.12 | 1.36 | 133.24 | 124.55 |

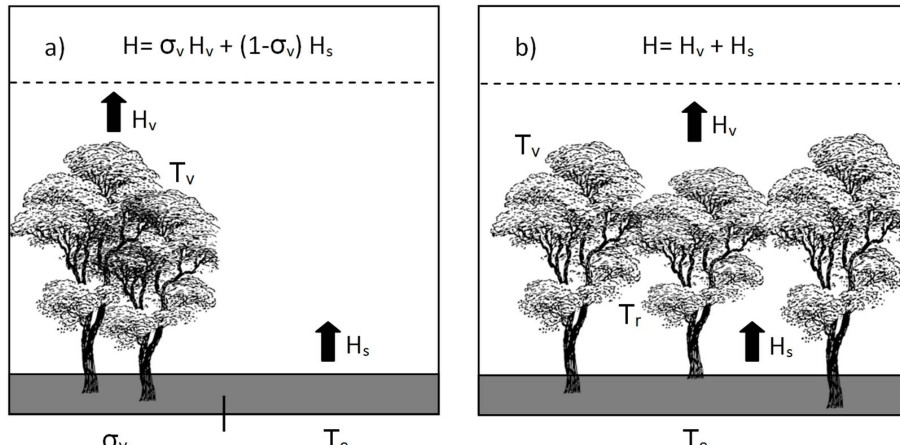

**Figure 1.** The representation of vegetation in (a) MOSES, where vegetation is beside bare ground and in (b) CABLE, where vegetation is above the ground. The mean grid heat flux, H, in MOSES is a sum of the fluxes weighted by the tile fractions e.g. vegetation fraction, $\sigma_v$. In CABLE, H is a sum of canopy, $H_v$, and soil fluxes, $H_s$. The vegetation, soil and radiative tempeartures are $T_v$, $T_s$ and $T_r$ respectively.

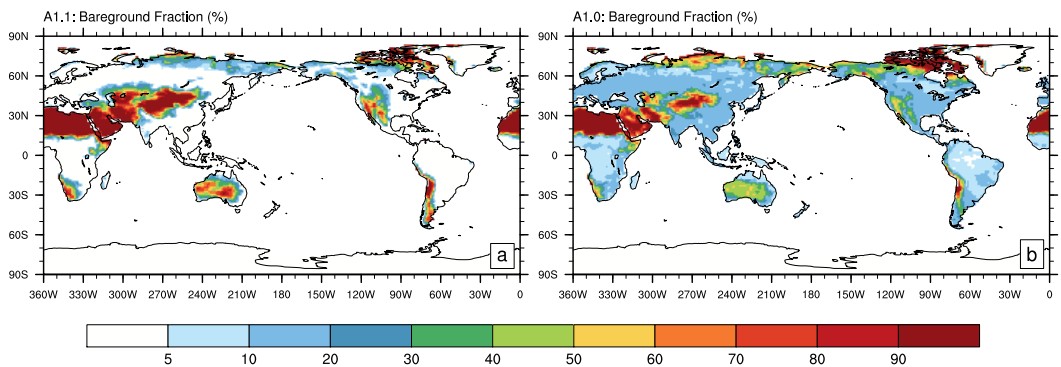

**Figure 2.** Percentage grid-cell coverage of the bare ground surface type for (a) ACCESS1.1 and (b) ACCESS1.0.

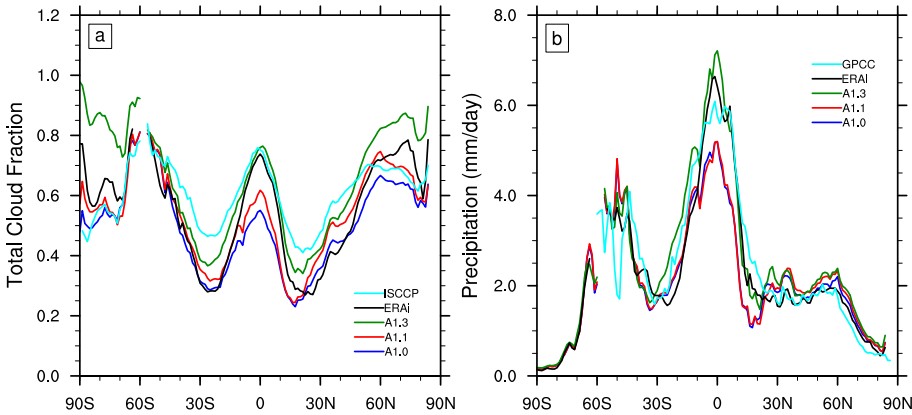

**Figure 3.** Zonal land-only average (a) total cloud fraction and (b) precipitation $(\mathrm{mm\,day^{-1}})$ for ISCCP (1984-2003)/GPCC(1979-1998), ERA-Interim (1979-1998), ACCESS1.0 (1979-1998), ACCESS1.1 (1979-1998) and ACCESS1.3 (1979-2005).

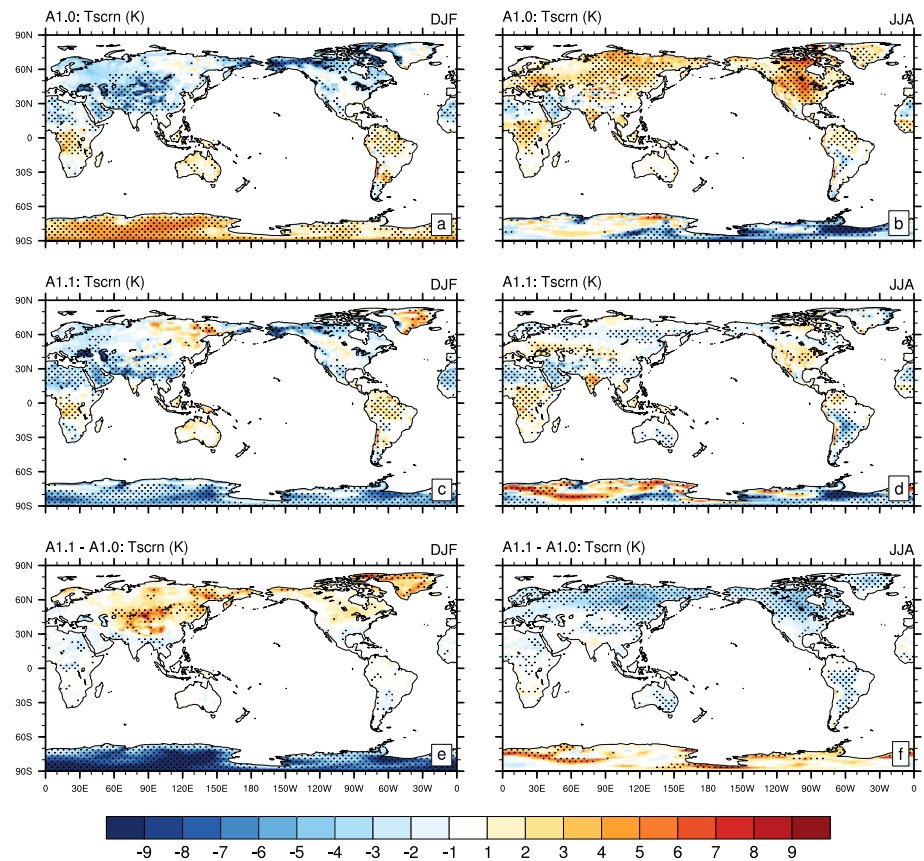

**Figure 4.** Seasonal mean screen temperature biases (°C) for ACCESS1.0 (a,b) and ACCESS1.1 (c,d) AMIP simulation evaluated against ERA-Interim analysis for DJF (left column) and JJA (right column). The model screen temperature difference, ACCESS1.1 minus ACCESS1.0, is shown in (e,f). Areas of statistical significance based on the modified t-test are shown in all panels via stippling.

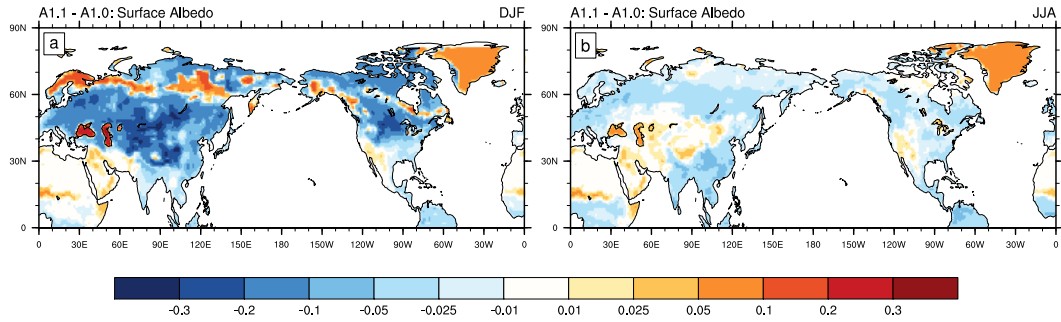

**Figure 5.** Seasonal mean surface albedo difference between ACCESS1.1 and ACCESS1.0 for (a) DJF and (b) JJA.

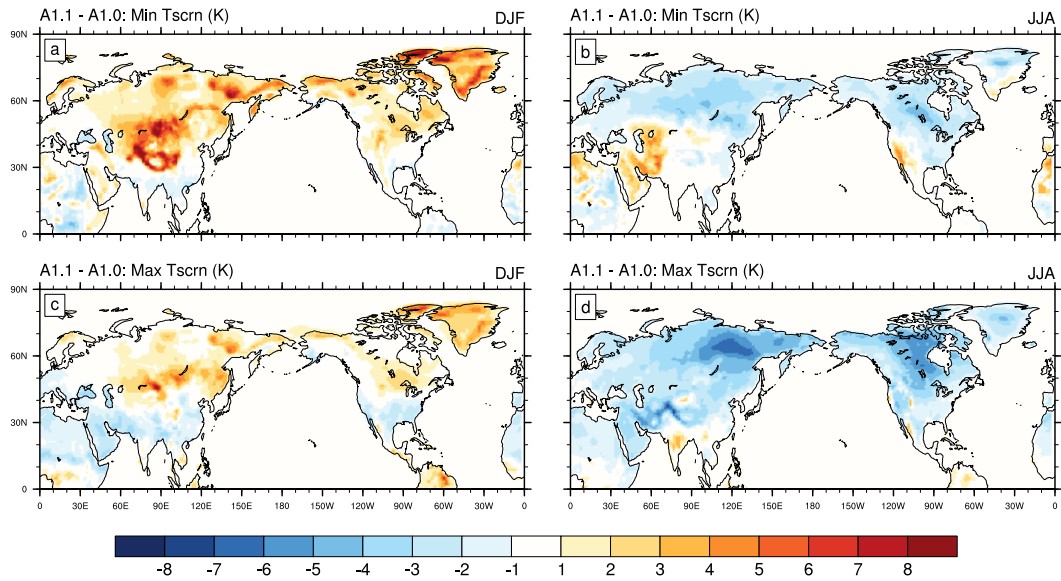

**Figure 6.** Northern hemisphere seasonal minimum and maximum screen temperature (K) difference between ACCESS1.1 and ACCESS1.0 for DJF and JJA.

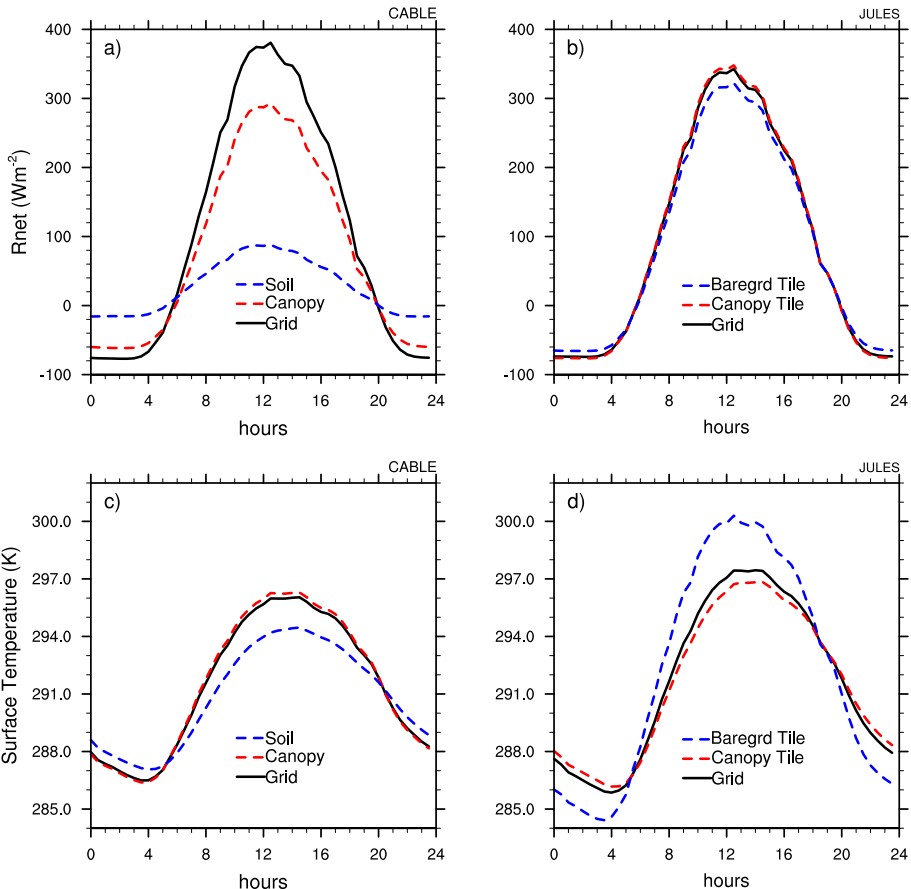

**Figure 7.** Offline simulation of Hyytiälä, 2002-2005. July mean diurnal cycles of net radiation $(\mathrm{W\,m^{-2}})$ for CABLE (a) and JULES (b) and temperature (K) for CABLE (c) and JULES (d) for grid-cell (black), vegetation (red) and soil (blue) for CABLE and grid-cell (black), vegetation (red) and bare ground (blue) tiles for JULES.

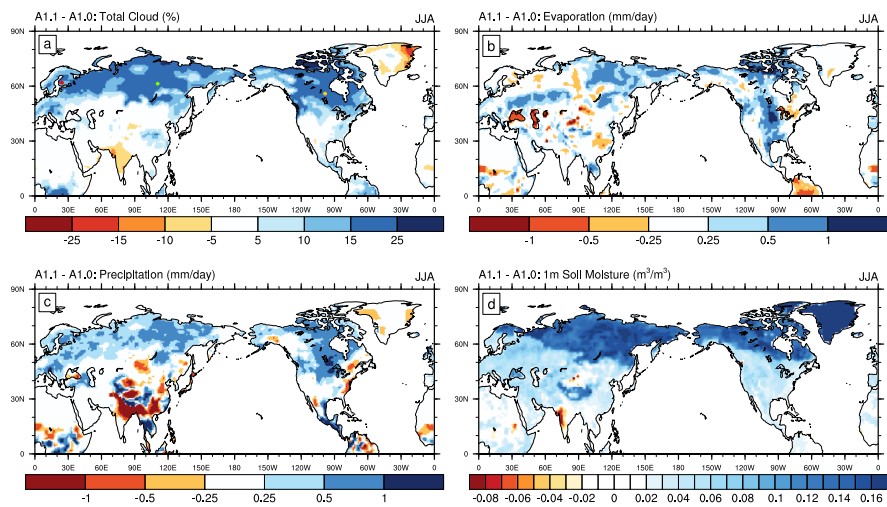

**Figure 8.** Seasonal mean difference in (a) total cloud fraction, (b) evaporation (mm/day), (c) precipitation (mm/day) and (d) 1 metre soil moisture ($m^3/m^3$) between ACCESS1.1 and ACCESS1.0 for JJA. Boreas (Canada), East Siberia (Russia) and Hyytiälä (Finland) marked as yellow, green and red dots, respectively, in panel (a).

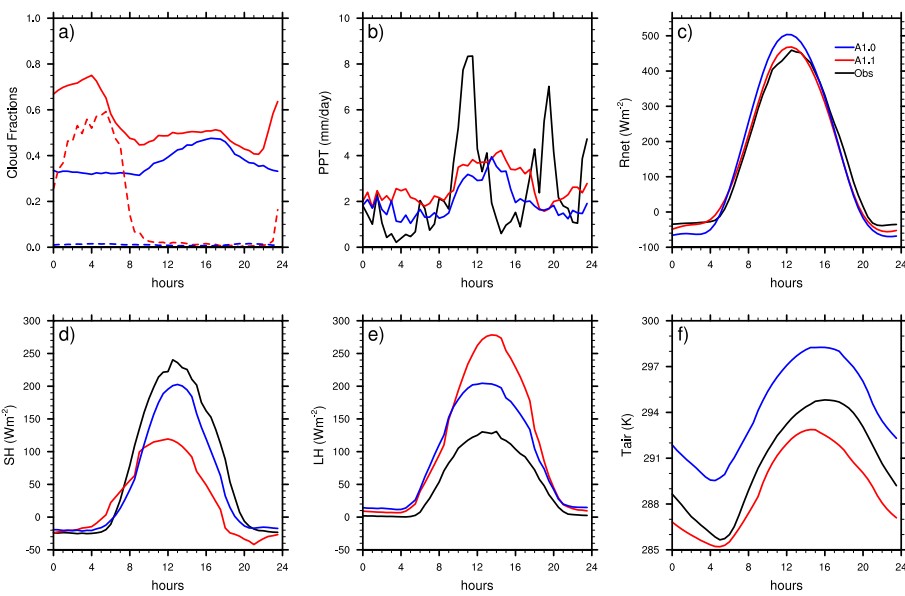

**Figure 9.** July diurnal cycles of (a) Total Cloud Fraction (solid) and (1yr average only) Very Low cloud Fraction (dash), (b) Precipitation (PPT, mm/day), (c) net radiation (Rnet, $Wm^{-2}$), (d) sensible heat (SH, $Wm^{-2}$), (e) latent heat (LH, $Wm^{-2}$), and (f) Screen/Air Temperature (Tair, K) for Boreas. Observations in black, ACCESS1.0 in blue and ACCESS1.1 in red.

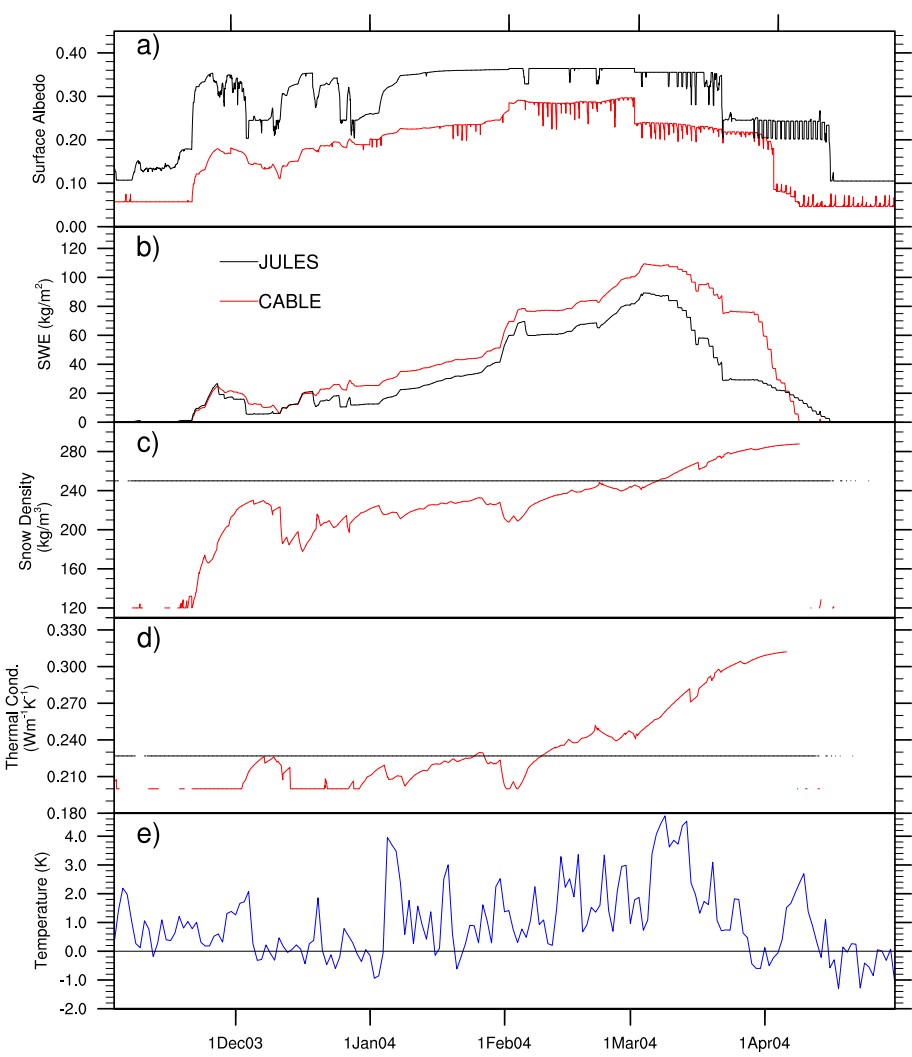

**Figure 10.** Offline simulation of 2003/2004 winter snow processes in Hyytiälä for CABLE (red) and JULES (black); (a) surface albedo, (b) snow water equivalent (SWE) in kg/m$^2$, (c) snow density in kg/m$^3$, (d) thermal conductivity in Wm$^{-1}$K$^{-1}$ and e) the mean daily temperature difference, CABLE minus JULES, in K.

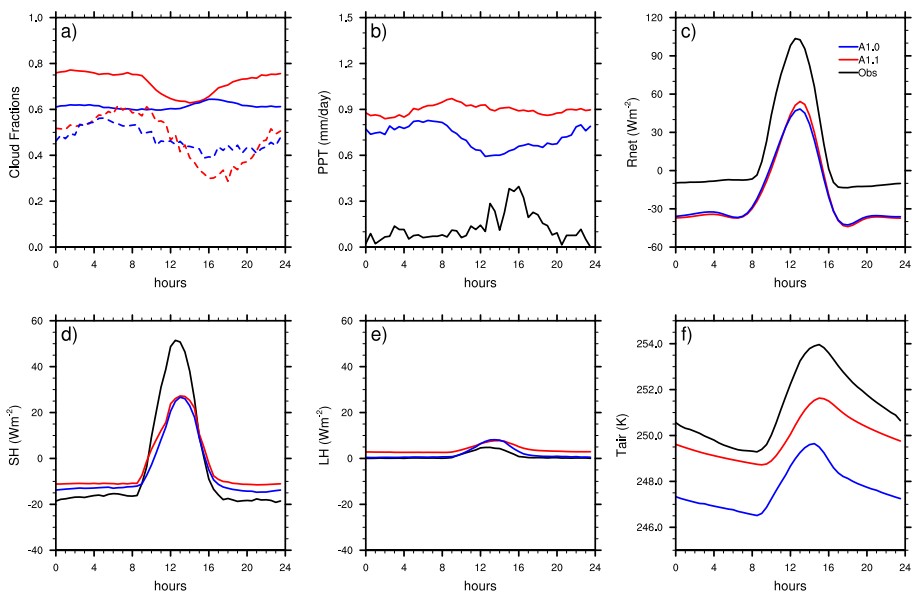

**Figure 11.** January diurnal cycles of (a) Total Cloud Fraction (solid) and (1yr average only) Very Low cloud Fraction (dash), (b) Precipitation (PPT, mm/day), (c) net radiation (Rnet, Wm$^{-2}$), (d) sensible heat (SH, Wm$^{-2}$), (e) latent heat (LH, Wm$^{-2}$), and (f) Screen/Air Temperature (Tair, K) for Boreas (55.88$^{\circ}$N, -98.48$^{\circ}$E). Observations in black, ACCESS1.0 in blue and ACCESS1.1 in red.

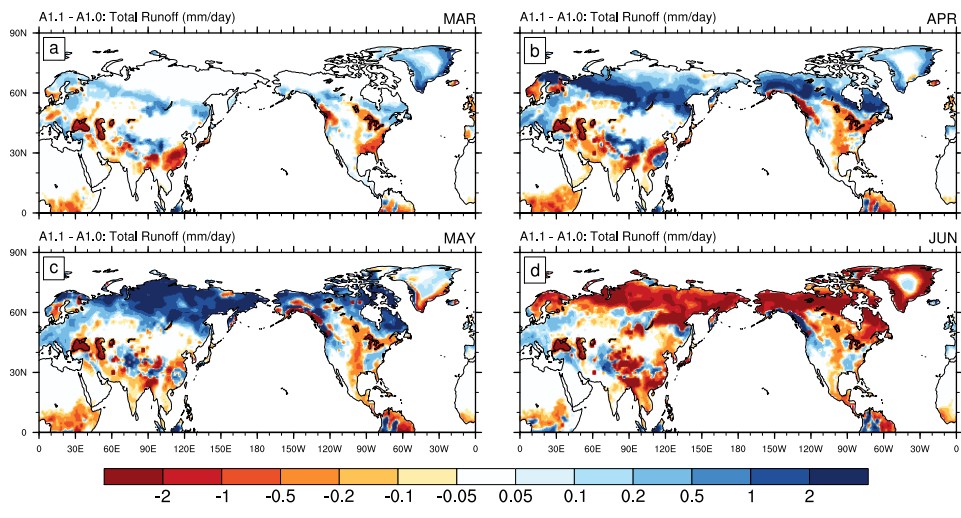

**Figure 12.** Monthly mean total runoff (mm/day) difference between ACCESS1.1 and ACCESS1.0 AMIP simulation for (a) March, (b) April, (c) May, and (d) June.