# Peer review of "The impact of changing the land surface scheme in ACCESS(v1.0/1.1) on the surface climatology"

_Geoscientific Model Development, 2016_

## Referee Comment (RC1) · Anonymous Referee #1 · 1 Apr 2016

General comments:

This paper try to analyze the impact of changing the MOSES land surface scheme by the CABLE land surface model in the ACCESS climate model. The introduction is very short and any references/discussion about land/atmosphere coupling processes (c.f. Betts et al. 1996; Betts 2009) were done. Only 2 references on previous works analyzing land surface-atmosphere interactions were done without any discussion. In the section 2, differences between the both land surface modules are relatively well described even if some details are missing. The section 3 presents the experimental design of the study. There is only two sentences that present the observations used to evaluate the model, that underline the poor scientific quality of this study. ERA Interim
reanalysis alone cannot be used as observations. There is many "real" observations available to evaluate the model. The section 4 presents the main results. The text is very descriptive and generally boring even if the fact to use off-line runs to explain some in-line behaviours is a very good idea. Any tests of significance is done for all differences model versus observations and model versus model shown in this manuscript. The conclusion is well written and brings into focus the qualities and the defaults of this study. After having hesitated for a long time between rejected this paper or reconsidered it after major revisions, I think that this paper must be largely improved and it deserves a chance.

Specific comments:

P.1 - 2 : The introduction did not give the readers a sense of the state of the art, e.g., what are the land/atmosphere coupling processes, and which ones are important or not ? Did anyone else in the community attempt to analyze land/atmosphere coupling for global climate simulations? Only one sentence to sum up the work of Koster (2004) or Seneviratne et al. (2010) is not enough.

P.1, l.16 : I am not sure that LSM is a "key" component of a climate model, it is an important component but the key component is the atmospheric model (or the oceanic model).

P.3, l.25 : What is the value for Cs or how it is computed?

P.3, l.27 : What is the value for fr or how it is computed ?

P.4, l.19 : What is the value for c in the exponential or how it is computed ?

P.4, l.20 and l.30: Are you sure that CABLE "has a more complex representation" of canopy or displacement height than many LSMs? If so, prove it and explain the main differences with "many other LSMs".

P.5, l.17 : "MOSES does account for this heat exchange." These sentences are not clear to me.

P.5, l.31 : The description of snow modules are very short while in the paper you access that "Warmer winter temperatures simulated by ACCESS1.1 over the snow covered areas of mid and high latitudes are attributed to differences in the snow parameterization in CABLE compared with MOSES." Please provide more details. Does CABLE "has a more complex representation" of snow processes than many other LSMs ?

P.6, l.27 : LAI data are not the same for MOSES and CABLE. How you can be sure that most of the impacts in summer are not due directly to this difference rather than in the differences in land surface physics ? This fact must be addressed in order to improve the scientific quality of this paper. Normally, in this paper, the same LAI should be used in both models, perhaps by introducing an intermediate simulation (MOSES with LAI from CABLE or inversely).

P.7, l. : This is the most important negative point of this work. Any "real" observation was used. For continental precipitation, there is product : GPCC (certainly the best), GPCP, TRIMM (over tropics), etc... Try to use these products in your comparison. The same is true for temperature or cloud: CRU data gives Tasmin/Tasmax/Tasmean that allow to evaluate the diurnal cycle ; CERES, MODIS, etc... for cloud (see Pincus et al. 2012)

P.8 and after : The result and figure part are too descriptive and not enough scientific. Tests of significance (for example T-Test) must be done on the differences model vs obs as well as for model vs model. On figure, generally, the pattern where the differences are significant is shown with dotted panel. Without that, this article can not be accepted. Please only comment where the differences are significant statistically.

References :

Betts, A. K., J. H. Ball, A. C. M. Beljaars, M. J. Miller, and P. A. Viterbo (1996), The land surface-atmosphere interaction: A review based on observational and global modeling perspectives, J. Geophys. Res., 101(D3), 7209–7225, doi:10.1029/95JD02135.

Betts, A. K. (2009), Land-Surface-Atmosphere Coupling in Observations and Models, J. Adv. Model. Earth Syst., 1, 4, doi:10.3894/JAMES.2009.1.4.

Pincus, R., S. Platnick, S. A. Ackerman, R. S. Hemler, and R. J. P. Hofmann, 2012: Reconciling simulated and observed views of clouds: MODIS, ISCCP, and the limits of instrument simulators. J. Climate, 25, 4699–4720

———————————————

---

## Referee Comment (RC2) · Anonymous Referee #2 · 21 Apr 2016

**Review of: "The impact of surface climatology from changing the land surface scheme in the ACCESS(v1.0/1.1) climate model"**

The paper describes the impact of using CABLE rather than MOSES (the default LSM) within the ACCESS model system. This is a highly relevant paper as ACCESS is a widely used model within the Australian climate science community. The paper is well written, and the physical mechanisms behind the changes are very well explored and explained. The description of the model differences between CABLE and MOSES is also very valuable. Hence the paper is suitable for final publication in GMD with minor revisions. The following comments may help improve the manuscript.

**Major comments:**
 Title: Suggest changing to "The impact of changing the land surface scheme in ACCESS (v1.0/1.1) on the surface climatology" – Reads better

Abstract:
It is mentioned that CABLE results in a warmer winter and cooler summer in the NH, but no mention is made if this improves or degrades the bias?

The abstract should have a sentence or two, describing the overall effect of replacing MOSES with CABLE. The dynamics are very well explained, but it would leave a non-expert reader wondering: "was it worth the effort to replace MOSES with CABLE?" Although this is not the aim of this paper, an ACCESS user who is not an LSM-expert should be able to use the abstract as a guide to decide if they want to use CABLE versus MOSES. A few sentences could make this clearer.

Section 2.1: It is mentioned that subsurface tiling is used in CABLE. Would be useful to provide information if the maximum number of tiles per grid-cell is user-specified, or automatically computed?

In the same section, would it be possible to tabulate the differences between CABLE and MOSES in table format? That would be a useful summary for ACCESS users to be able to refer to.

Section 3.3: It is fine that you are using the offline simulations to focus on model behaviour rather than reproduce the online results, but a reader would be left wondering why you did not use same versions for the offline simulations.

In the same section, line 21, provide a CABLE version, and perhaps state the UM version with the different atmospheric physics.

Figure 4: The improved bias during JJA with ACCESS1.1 (CABLE) as compared to 1.0 (MOSES) over NA and northern Europe could be mentioned in the abstract.

The discussion of the physical mechanisms behind the differences between CABLE and MOSES in ACCESS is very thorough and convincing. The conclusion could use a few sentences on "what this all means". It seems to me that CABLE in

ACCESS, with it's more realistic method of energy portioning etc, is an important step forwards in ACCESS development. I suggest a paragraph, aimed at a non-LSM experts, which paints the broader picture.

There should be some rational for the use of Era-Interim. This has been raised by the first reviewer. Perhaps the authors should clarify that they use ERA-Interim such that they can investigate L-A feedbacks in a consistent manner? i.e., one can make inferences about temp, precip, cloud feedbacks using Era-Interim, but this is harder to do using pure observational data-sets. The aim here is to investigate the feedbacks, and the use of Era-Interim seems appropriate to me.

The first reviewer has also commented on the lack of statistics used in this paper. I do not think the use of statistical significance testing would add much to this paper. The aim is to investigate the physical mechanisms, as the authors have carried out. So, I would disagree with the first reviewer on this point.

**Editorial comments:**
Page 1, line 8, replace "placement of canopy" with "placement of the canopy".
Page 1, line 11, replace "lowers diurnally" with "lowers the diurnally"
Page 2, line 11, replace "while (Kowalczky et al. 2013)" with "while Kowalczky et al. (2013)".
Page 2, line 24, the "HadGEM2 Development Team: et al. (2011)" reference seems strange?
Page 2, line 28, suggest to add/provide some references after "interpret the results from ACCESS1.3".
Page 3, line 15, replace "structural placing" with "the structural placement".
Figure 1 caption: provide descriptions of H, Hv, Hs, sigma_v etc.
Page 3, line 24, L does not appear in Eq. 1
Page 4, line 2, Fig. 2 should be in brackets? Or "as shown in Fig. 2", and elsewhere in the manuscript, e.g., line 4.
Page 4, line 4, what "many other LSMs", should provide references. Are you referring to CLM, NOAH, ORCHIDEE etc? or is this a broad statement?
Page4, line 8 – Should note that it is possible to parameterize snow-free albedo in CABLE as described in Kala et al. (2014) (www.geosci-model-dev.net/7/2121/2014/), but this is yet to be tested coupled to ACCESS and not usually activated.

Page 9, line 1, replace "In boreal summer" with "During the Boreal summer".

Page 10, line 12, replace "giving lower surface albedo" with "simulating lower surface albedo". The phrase "CABLE gives…." is use a lot through the manuscript. Suggest to use "simulates" instead.

Page 9, last line, impacts "the" overall

Page 10, line 29, "The" Boreas grid-cell

Page 15, last line. Recent work would suggest that stomatal opening to CO2 is equally important: Kala et al. (2015):

---

## Author Comment (AC2) · 13 May 2016

We thank the reviewer for their comments. Each comment is addressed below with the original review in italics and our responses in normal font.

*The paper describes the impact of using CABLE rather than MOSES (the default LSM) within the ACCESS model system. This is a highly relevant paper as ACCESS is a widely used model within the Australian climate science community. The paper is well written, and the physical mechanisms behind the changes are very well explored and explained. The description of the model differences between CABLE and MOSES is also very valuable. Hence the paper is suitable for final publication in GMD with minor revisions. The following comments may help improve the manuscript.*

*Major comments:*
*Title: Suggest changing to "The impact of changing the land surface scheme in ACCESS (v1.0/1.1) on the surface climatology" – Reads better*

We will revise the title so it is easier to read.

*Abstract:*
*It is mentioned that CABLE results in a warmer winter and cooler summer in the NH, but no mention is made if this improves or degrades the bias?*

Significance testing as requested by reviewer 1 suggests that A1.1 (CABLE) has smaller bias in summer than A1.0 (MOSES) while the relative model performance cannot be distinguished in winter. This will now be noted in the abstract.

*The abstract should have a sentence or two, describing the overall effect of replacing MOSES with CABLE. The dynamics are very well explained, but it would leave a non-expert reader wondering: "was it worth the effort to replace MOSES with CABLE?" Although this is not the aim of this paper, an ACCESS user who is not an LSM-expert should be able to use the abstract as a guide to decide if they want to use CABLE versus MOSES. A few sentences could make this clearer.*

A summary statement will be added to the end of the abstract indicating how the extra complexity in CABLE benefits the ACCESS simulation.

*Section 2.1: It is mentioned that subsurface tiling is used in CABLE. Would be useful to provide information if the maximum number of tiles per grid-cell is user-specified, or automatically computed?*

We will add this information is section 3.1 paragraph 4.

*In the same section, would it be possible to tabulate the differences between CABLE and MOSES in table format? That would be a useful summary for ACCESS users to be able to refer to.*

We now will include a table referenced from section 2.2 to summarise the differences including further details about the snow scheme as requested by reviewer 1.

*Section 3.3: It is fine that you are using the offline simulations to focus on model behaviour rather than reproduce the online results, but a reader would be left wondering why you did not use same versions for the offline simulations.*

Earlier versions of the code were not setup to easily switch between offline and online simulations. This information will be added in section 3.3.

*In the same section, line 21, provide a CABLE version, and perhaps state the UM version with the different atmospheric physics.*

We will add the versions and appropriate reference in manuscript.

*Figure 4: The improved bias during JJA with ACCESS1.1 (CABLE) as compared to 1.0 (MOSES) over NA and northern Europe could be mentioned in the abstract.*

As noted above this finding will be added to the abstract.

*The discussion of the physical mechanisms behind the differences between CABLE and MOSES in ACCESS is very thorough and convincing. The conclusion could use a few sentences on "what this all means". It seems to me that CABLE in ACCESS, with its more realistic method of energy portioning etc, is an important step forwards in ACCESS development. I suggest a paragraph, aimed at a non-LSM experts, which paints the broader picture.*

We agree that the paper would benefit from an additional paragraph in the conclusions as suggested by the reviewer. We will focus on how the unique features of CABLE contribute to ACCESS development.

*There should be some rational for the use of Era-Interim. This has been raised by the first reviewer. Perhaps the authors should clarify that they use ERA-Interim such that they can investigate L-A feedbacks in a consistent manner? i.e., one can make inferences about temp, precip, cloud feedbacks using Era-Interim, but this is harder to do using pure observational data-sets. The aim here is to investigate the feedbacks, and the use of Era-Interim seems appropriate to me.*

As suggested by reviewer 1 we have now confirmed our model-obs comparisons against other datasets than just Era Interim, noting also that much of the paper focuses on model-model comparisons for which observations are not relevant. The manuscript will be updated accordingly particularly section 3.2 and discussion of fig 3 and 4 a-d.

*The first reviewer has also commented on the lack of statistics used in this paper.*
*I do not think the use of statistical significance testing would add much to this paper. The aim is to investigate the physical mechanisms, as the authors have carried out. So, I would disagree with the first reviewer on this point.*

We have now performed significance testing for model-model and model-obs differences. The model-model differences are significant almost everywhere and we agree with the reviewer that this would not add much to the paper. The significance of model-obs differences may be helpful for commenting on whether one model or the other gives smaller biases compared to obs/reanalysis.

*Editorial comments:*
*Page 1, line 8, replace "placement of canopy" with "placement of the canopy".*

We will change this in the manuscript.

*Page 1, line 11, replace "lowers diurnally" with "lowers the diurnally"*

We will change this in the manuscript.

*Page 2, line 11, replace "while (Kowalczky et al. 2013)" with "while Kowalczky et al. (2013)".*

We will change this in the manuscript.

*Page 2, line 24, the "HadGEM2 Development Team: et al. (2011)" reference seems strange?*

We believe this follows GMD style guidelines.

*Page 2, line 28, suggest to add/provide some references after "interpret the results from ACCESS1.3".*

We will add references to Bi et al 2013 and Kowalczyk et al 2013

*Page 3, line 15, replace "structural placing" with "the structural placement".*

We will change this in the manuscript.

*Figure 1 caption: provide descriptions of H, Hv, Hs, sigma_v etc.*

We will change this in the manuscript.

*Page 3, line 24, L does not appear in Eq. 1*

L is in equation 1 as a part of LE. This will be made clearer in the definition of the variables in the equation.

*Page 4, line 2, Fig. 2 should be in brackets? Or "as shown in Fig. 2", and elsewhere in the manuscript, e.g., line 4.*

We will change this in the manuscript.

*Page 4, line 4, what "many other LSMs", should provide references. Are you referring to CLM, NOAH, ORCHIDEE etc? or is this a broad statement?*

This is a broad statement which we now clarify by noting that these other LSMs tend to use conventional rough wall boundary theory, with parameters that are a constant fraction of canopy height.

*Page4, line 8 – Should note that it is possible to parameterize snow-free albedo in CABLE as described in Kala et al. (2014) (www.geosci-modeldev.net/7/2121/2014/), but this is yet to be tested coupled to ACCESS and not usually activated.*

We will add this information in section 2.2.

*Page 9, line 1, replace "In boreal summer" with "During the Boreal summer".*

We will change this in the manuscript.

*Page 10, line 12, replace "giving lower surface albedo" with "simulating lower surface albedo". The phrase "CABLE gives…." is use a lot through the manuscript. Suggest to use "simulates" instead.*

We will change this in the manuscript.

*Page 9, last line, impacts "the" overall*

We will add this in the manuscript.

*Page 10, line 29, "The" Boreas grid-cell*

We will add this in the manuscript.

*Page 15, last line. Recent work would suggest that stomatal opening to CO2 is equally important: Kala et al. (2015): http://www.nature.com/articles/srep23418*

We will add this in the manuscript.

---

## Author Response (AR1)

We thank the reviewer for their comments. Each comment is addressed below with the original review in italics, our responses in normal font and changes to the manuscript in **bold**.

*General comments:*
*This paper try to analyze the impact of changing the MOSES land surface scheme by the CABLE land surface model in the ACCESS climate model. The introduction is very short and any references/discussion about land/atmosphere coupling processes (c.f. Betts et al. 1996; Betts 2009) were done. Only 2 references on previous works analyzing land surface-atmosphere interactions were done without any discussion.*

We will provided extra detail in the introduction especially around the Betts 2009 paper and recent work by Hirsch et al **(p1, L23 – p2, L17)**.

*In the section 2, differences between the both land surface modules are relatively well described even if some details are missing.*

As requested by reviewer 2 we now include a table of differences including some further details about the snow scheme **(p4, L3, new Table 1, p23)**.

*The section 3 presents the experimental design of the study. There is only two sentences that present the observations used to evaluate the model, that underline the poor scientific quality of this study. ERA Interim reanalysis alone cannot be used as observations. There is many "real" observations available to evaluate the model.*

The main purpose of this paper was a comparative study of simulations with different land surface schemes. For this reason most of the results presented are model-model comparisons (and a process understanding of these model differences) rather than model-obs comparisons. Where we show zonal mean cloud and precipitation we will add an observational dataset and will now show land only zonal means to be compatible with the GPCC dataset **(p25, fig 3)**. We understand the limitation of only using ERA interim reanalysis but have now checked our model simulations against CRU temperatures as well. This will be noted in our comments on the relative bias of each model temperature compared to ERA Interim/obs **([p9, L8-20], [p10, L3-9])**.

*The section 4 presents the main results. The text is very descriptive and generally boring even if the fact to use off-line runs to explain some in-line behaviours is a very good idea.*

We will revise **section 4** to remove some unnecessary detail and to try to make it easier to read.

*Any tests of significance is done for all differences model versus observations and model versus model shown in this manuscript.*

See reply below about significance testing **(p9, L26 - p10, L11)**.

*The conclusion is well written and brings into focus the qualities and the defaults of this study. After having hesitated for a long time between rejected this paper or reconsidered it after major revisions, I think that this paper must be largely improved and it deserves a chance.*

We will endeavour to address both reviewer comments, noting that at times they had substantially different views on the relative importance of different parts of the paper and how it might be

improved. We believe that our changes will deal with all critical issues and will result in an improved manuscript.

*Specific comments:*
*P.1-2: The introduction did not give the readers a sense of the state of the art, e.g., what are the land/atmosphere coupling processes, and which ones are important or not? Did anyone else in the community attempt to analyze land/atmosphere coupling for global climate simulations? Only one sentence to sum up the work of Koster (2004) or Seneviratne et al. (2010) is not enough.*

As our response above we will extend the introduction to give more information about the cited work and to add some other relevant literature **(p1, L23 – p2, L17)**.

*P.1, l.16: I am not sure that LSM is a "key" component of a climate model, it is an important component but the key component is the atmospheric model (or the oceanic model).*

Sentence will be modified to note that LSM is one of the key components **(p1, L19)**.

*P.3, l.25: What is the value for Cs or how it is computed?*

Cs is a volumetric heat capacity calculated as the weighted sum of the heat capacity of dry soil, liquid and ice ($JK^{-1}m^{-2}$) and this will be added to the manuscript **(p4, L13)**.

*P.3, l.27: What is the value for fr or how it is computed?*

Fr is $1-e^{LAI/2}$ and this will be added to the manuscript **(p4, L18)**.

*P.4, l.19: What is the value for c in the exponential or how it is computed?*

C is an extinction coeffecient for beam radiation and black leaves and this will be added to the manuscript **(p5, L10)**.

*P.4, l.20 and l.30: Are you sure that CABLE "has a more complex representation" of canopy or displacement height than many LSMs? If so, prove it and explain the main differences with "many other LSMs".*

Most other LSM use conventional rough wall boundary layer theory and canopy parameters, e.g. displacement height, that are a constant fraction of canopy height. CABLE more complex representation is based on Raupach 1989, 1994 which are referenced in the paper. We will note in the text the simpler methods used by most other models **([p5, L12-13], [p5, L23-24])** but do not feel it is necessary to repeat all the detail on the canopy turbulent transport described in the Raupach references.

*P.5, l.17: "MOSES does account for this heat exchange." These sentences are not clear to me.*

This will be clarified - Moses calculates the advection of heat by moisture fluxes within the soil column **(p6, L13)**.

*P.5, l.31 : The description of snow modules are very short while in the paper you access that "Warmer winter temperatures simulated by ACCESS1.1 over the snow covered areas of mid and high latitudes are attributed to differences in the snow parameterization in CABLE compared with MOSES." Please*

*provide more details. Does CABLE "has a more complex representation" of snow processes than many other LSMs?*

Differences in the snow parameterisation will now be captured in the new table of CABLE-MOSES model differences which was requested by reviewer 2 **(p4, L3, new Table 1, p23)**.

*P.6, l.27: LAI data are not the same for MOSES and CABLE. How you can be sure that most of the impacts in summer are not due directly to this difference rather than in the differences in land surface physics? This fact must be addressed in order to improve the scientific quality of this paper. Normally, in this paper, the same LAI should be used in both models, perhaps by introducing an intermediate simulation (MOSES with LAI from CABLE or inversely).*

We have now performed a simulation in which ACCESS1.0 (MOSES) is run with ACCESS1.1 (CABLE's) LAI **(p8, L19-21)**. The change in LAI has very little impact on the simulation. The figure below shows the difference between the sensitivity test and the original A1.0 and should be compared with Fig 4 e/f from the paper. The very small differences shown do not explain the A1.1-A1.0 differences from Fig 4. Appropriate comments will be added to the manuscript when explaining the temperature differences found for the Northern continent in summer **(p10, L34 - p11, L4)** and winter **(p13, L25-32)**.

[Figure]

*P.7, l.: This is the most important negative point of this work. Any "real" observation was used. For continental precipitation, there is product: GPCC (certainly the best), GPCP, TRIMM (over tropics), etc. Try to use these products in your comparison. The same is true for temperature or cloud: CRU data gives Tasmin/Tasmax/Tasmean that allow to evaluate the diurnal cycle; CERES, MODIS, etc. for cloud (see Pincus et al. 2012)*

As noted above we will add some extra comparisons with other observational products where the paper currently focusses on model differences from observations (Fig 3 & 4 a-d) **(p8, L5-11) (ISCCP/GPCC -[p9, L8-20], CRU-[p10, L3-9])**. The rest of the paper focusses on model-model differences and a process understanding of these differences and hence additional comparisons to observations are less relevant for that analysis.

*P.8 and after: The result and figure part are too descriptive and not enough scientific.*
*Tests of significance (for example T-Test) must be done on the differences model vs obs as well as for model vs model. On figure, generally, the pattern where the differences are significant is shown with dotted panel. Without that, this article cannot be accepted. Please only comment where the differences are significant statistically.*

Original Author Response, 13 May 2016: In general significance testing shows that model-model differences are significant almost everywhere (see figure below for seasonal mean temperature) while the significance of model-obs differences is both seasonally and model dependent. Model-obs significance testing is consequently useful for determining if one model better simulates observations than the other model. Testing against both CRU and ERA Interim temperatures suggests that the models can't be differentiated in DJF but that A1.1 (CABLE) produces the better simulation of temperature in JJA **(p9, L26 - p10, L11)**. T-test stippling will be added to Fig 4 a-d **(p26)** but we will not add stippling to the other figures of model-model differences since it would be required almost everywhere and tends to obscure important features of these figures.

[Figure]

Additional Author Response, 2 June 2016: To follow-up the request from reviewer 1 for significance testing, we have now also explored the use of the modified t-test (Zwiers and von Storch, 1995). This accounts for temporal correlation and the relatively small sample size which we have available (~20 years). Compared to the standard t-test, the modified t-test generally gives less widespread areas of significance, particularly for the model-model differences in northern winter. Based on this new testing, we propose showing significance stippling for both the model-ERAi temperature biases and the model-model differences in Fig 4, as these are the critical results that we are trying to explain.

Since the model-model differences shown in the remainder of the paper are primarily for explaining the temperature differences, we feel that it is not important to include the t-test stippling on those figures.

Zwiers, F. W. and von Storch, H.: Taking Serial Correlation in Account inTests of the Mean, J. Climate, 8, 336-351, 1995.

We thank the reviewer for their comments. Each comment is addressed below with the original review in italics, our responses in normal font and changes to the manuscript in **bold**.

*The paper describes the impact of using CABLE rather than MOSES (the default LSM) within the ACCESS model system. This is a highly relevant paper as ACCESS is a widely used model within the Australian climate science community. The paper is well written, and the physical mechanisms behind the changes are very well explored and explained. The description of the model differences between CABLE and MOSES is also very valuable. Hence the paper is suitable for final publication in GMD with minor revisions. The following comments may help improve the manuscript.*

*Major comments:*
*Title: Suggest changing to "The impact of changing the land surface scheme in ACCESS (v1.0/1.1) on the surface climatology" – Reads better*

We will revise the title so it is easier to read **(p1, title)**.

*Abstract:*
*It is mentioned that CABLE results in a warmer winter and cooler summer in the NH, but no mention is made if this improves or degrades the bias?*

Significance testing as requested by reviewer 1 suggests that A1.1 (CABLE) has smaller bias in summer than A1.0 (MOSES) while the relative model performance cannot be distinguished in winter. This will now be noted in the abstract **(p1, L6-7)**.

*The abstract should have a sentence or two, describing the overall effect of replacing MOSES with CABLE. The dynamics are very well explained, but it would leave a non-expert reader wondering: "was it worth the effort to replace MOSES with CABLE?" Although this is not the aim of this paper, an ACCESS user who is not an LSM-expert should be able to use the abstract as a guide to decide if they want to use CABLE versus MOSES. A few sentences could make this clearer.*

A summary statement will be added to the end of the abstract indicating how the extra complexity in CABLE benefits the ACCESS simulation **(p1, L14-16)**.

*Section 2.1: It is mentioned that subsurface tiling is used in CABLE. Would be useful to provide information if the maximum number of tiles per grid-cell is user-specified, or automatically computed?*

We will add this information is section 3.1, paragraph 4 **(p7, line 19-20)**.

*In the same section, would it be possible to tabulate the differences between CABLE and MOSES in table format? That would be a useful summary for ACCESS users to be able to refer to.*

We now will include a table referenced from section 2.2 **(p4, L3)** to summarise the differences including further details about the snow scheme as requested by reviewer 1.

*Section 3.3: It is fine that you are using the offline simulations to focus on model behaviour rather than reproduce the online results, but a reader would be left wondering why you did not use same versions for the offline simulations.*

Earlier versions of the code were not setup to easily switch between offline and online simulations. This information will be added in section 3.3 **(p8, L26)**.

*In the same section, line 21, provide a CABLE version, and perhaps state the UM version with the different atmospheric physics.*

We will add the versions and appropriate reference in manuscript **(p8, L21-23)**.

*Figure 4: The improved bias during JJA with ACCESS1.1 (CABLE) as compared to 1.0 (MOSES) over NA and northern Europe could be mentioned in the abstract.*

As noted above this finding will be added to the abstract **(p1, L6-7)**.

*The discussion of the physical mechanisms behind the differences between CABLE and MOSES in ACCESS is very thorough and convincing. The conclusion could use a few sentences on "what this all means". It seems to me that CABLE in ACCESS, with its more realistic method of energy portioning etc, is an important step forwards in ACCESS development. I suggest a paragraph, aimed at a non-LSM experts, which paints the broader picture.*

We agree that the paper would benefit from an additional paragraph in the conclusions as suggested by the reviewer. We will focus on how the unique features of CABLE contribute to ACCESS development **(p16, L25-34)**.

*There should be some rational for the use of Era-Interim. This has been raised by the first reviewer. Perhaps the authors should clarify that they use ERA-Interim such that they can investigate L-A feedbacks in a consistent manner? i.e., one can make inferences about temp, precip, cloud feedbacks using Era-Interim, but this is harder to do using pure observational data-sets. The aim here is to investigate the feedbacks, and the use of Era-Interim seems appropriate to me.*

As suggested by reviewer 1 we have now confirmed our model-obs comparisons against other datasets than just Era Interim, noting also that much of the paper focuses on model-model comparisons for which observations are not relevant. The manuscript will be updated accordingly particularly section 3.2 **(p8, L5-11)** and discussion of fig 3 and 4 a-d **(p9, L25-p10, L11)**.

*The first reviewer has also commented on the lack of statistics used in this paper.*
*I do not think the use of statistical significance testing would add much to this paper. The aim is to investigate the physical mechanisms, as the authors have carried out. So, I would disagree with the first reviewer on this point.*

We have now performed significance testing for model-model and model-obs differences. The model-model differences are significant almost everywhere and we agree with the reviewer that this would not add much to the paper. The significance of model-obs differences may be helpful for commenting on whether one model or the other gives smaller biases compared to obs/reanalysis.

As per our additional author comment to reviewer 1 (2 June 2016) using the modified t-test generally gives reduced regions of significance. Consequently we have decided to include significance stippling on all panels of Figure 4 but don't think it necessary to show on the other model-model difference plots as their primary purpose is for explaining the temperature differences as suggested by this reviewer.

*Editorial comments:*

*Page 1, line 8, replace "placement of canopy" with "placement of the canopy".*

We will change this in the manuscript **(p1, L8)**.

*Page 1, line 11, replace "lowers diurnally" with "lowers the diurnally"*

We will change this in the manuscript **(p1, L11)**.

*Page 2, line 11, replace "while (Kowalczky et al. 2013)" with "while Kowalczky et al. (2013)".*

We will change this in the manuscript **(p2, L31)**.

*Page 2, line 24, the "HadGEM2 Development Team: et al. (2011)" reference seems strange?*

We believe this follows GMD style guidelines **(p3, L9)**.

*Page 2, line 28, suggest to add/provide some references after "interpret the results from ACCESS1.3".*

We will add references to Bi et al 2013 and Kowalczyk et al 2013 **(p3, L13)**.

*Page 3, line 15, replace "structural placing" with "the structural placement".*

We will change this in the manuscript **(p4, L2)**.

*Figure 1 caption: provide descriptions of H, Hv, Hs, sigma_v etc.*

We will change this in the manuscript **(p24)**.

*Page 3, line 24, L does not appear in Eq. 1*

L is in equation 1 as a part of LE. This will be made clearer in the definition of the variables in the equation **(p4, L12-13)**.

*Page 4, line 2, Fig. 2 should be in brackets? Or "as shown in Fig. 2", and elsewhere in the manuscript, e.g., line 4.*

We will change this in the manuscript **([p4, L22], [p4, L24], [p12, L14], [p14, L12], [p14, L27], [p14, L31], [p15, L12])**.

*Page 4, line 4, what "many other LSMs", should provide references. Are you referring to CLM, NOAH, ORCHIDEE etc? or is this a broad statement?*

This is a broad statement which we now clarify by noting that these other LSMs tend to use conventional rough wall boundary theory, with parameters that are a constant fraction of canopy height **([p5, L12-13], [p5, L23-24])**.

*Page4, line 8 – Should note that it is possible to parameterize snow-free albedo in CABLE as described in Kala et al. (2014) (www.geosci-modeldev.net/7/2121/2014/), but this is yet to be tested coupled to ACCESS and not usually activated.*

We will add this information in section 2.2 **(p6, L2-4)**.

*Page 9, line 1, replace "In boreal summer" with "During the Boreal summer".*

We have decided to change this in the manuscript to "In JJA" to explicitly link the figure and text **(p10, L26)**.

*Page 10, line 12, replace "giving lower surface albedo" with "simulating lower surface albedo". The phrase "CABLE gives...." is use a lot through the manuscript. Suggest to use "simulates" instead.*

We will change this in the manuscript **(p10, L27)**.

*Page 9, last line, impacts "the" overall*

We will add this in the manuscript **(p11, L20)**.

*Page 10, line 29, "The" Boreas grid-cell*

We will add this in the manuscript **(p12, L13)**.

*Page 15, last line. Recent work would suggest that stomatal opening to CO2 is equally important: Kala et al. (2015): http://www.nature.com/articles/srep23418*

We will add this in the manuscript **(p16, L23-24)**.

[revised manuscript text omitted]